# An alternative splicing program promotes adipose tissue thermogenesis

Santiago Vernia[1†], Yvonne JK Edwards[1], Myoung Sook Han[1],
Julie Cavanagh-Kyros[1,2], Tamera Barrett[1,2], Jason K Kim[1], Roger J Davis[1,2]*

[1]Program in Molecular Medicine, University of Massachusetts Medical School, Worcester, United States; [2]Howard Hughes Medical Institute, Worcester, United States

**Abstract** Alternative pre-mRNA splicing expands the complexity of the transcriptome and controls isoform-specific gene expression. Whether alternative splicing contributes to metabolic regulation is largely unknown. Here we investigated the contribution of alternative splicing to the development of diet-induced obesity. We found that obesity-induced changes in adipocyte gene expression include alternative pre-mRNA splicing. Bioinformatics analysis associated part of this alternative splicing program with sequence specific NOVA splicing factors. This conclusion was confirmed by studies of mice with NOVA deficiency in adipocytes. Phenotypic analysis of the NOVA-deficient mice demonstrated increased adipose tissue thermogenesis and improved glycemia. We show that NOVA proteins mediate a splicing program that suppresses adipose tissue thermogenesis. Together, these data provide quantitative analysis of gene expression at exon-level resolution in obesity and identify a novel mechanism that contributes to the regulation of adipose tissue function and the maintenance of normal glycemia.

*For correspondence: roger. davis@umassmed.edu

Present address: †Genes and Metabolism Section, MRC Clinical Sciences Centre, Hammersmith Campus, Imperial College London, London, United Kingdom

## Introduction

Alternative pre-mRNA splicing is an important mechanism that increases the complexity of the transcriptome (*Pan et al., 2008*; *Wang et al., 2008*) and expands the diversity and function of the proteome (*Nilsen and Graveley, 2010*; *Yang et al., 2016*). Indeed, differences in pre-mRNA splicing can contribute to the specialization of different cell types within the body (*Vaquero-Garcia et al., 2016*). The regulation of alternative pre-mRNA splicing is therefore an important aspect of cellular differentiation. Indeed, processes that regulate pre-mRNA splicing represent potential mechanisms that can control cell function. Recent studies have identified changes in pre-mRNA splicing associated with autism (*Irimia et al., 2014*; *Weyn-Vanhentenryck et al., 2014*), cardiac hypertrophy (*Mirtschink et al., 2015*), embryonic stem cell re-programming (*Han et al., 2013a*), tumorigenesis (*Oltean and Bates, 2014*; *Hsu et al., 2015*; *Koh et al., 2015*), and the regulation of signal transduction pathways (*Gupta et al., 1996*; *Tournier et al., 1999*; *Maimon et al., 2014*; *Martinez et al., 2015*). Moreover, alternative pre-mRNA splicing associated with pathogenesis represents a tractable target for the development of new therapies (*Daguenet et al., 2015*).

The role of alternative pre-mRNA splicing in metabolism is unclear. We studied pre-mRNA splicing in adipocytes to investigate whether adipocyte function may be regulated by changes in pre-mRNA splicing. Adipose tissue is critically important for whole body metabolic regulation because it acts as both an endocrine organ and as a storage depot for triglyceride (*Rosen and Spiegelman, 2014*). Interestingly, both adipose tissue deficiency (lipodystrophy) and adipose tissue accumulation (obesity) are associated with the development of metabolic syndrome and pre-diabetes (*Grundy, 2015*). Moreover, the widespread incidence of human obesity represents a major risk

**eLife digest** The process of building a protein from the information encoded in a gene begins when the gene is copied to form a pre-messenger RNA molecule. This molecule is then edited to produce a final messenger RNA that is "translated" to form the protein. Different segments of the pre-messenger RNA molecule can be removed to create different messenger RNAs. This "alternative splicing" enables a single gene to produce multiple protein variants, allowing a diverse range of processes to be performed by cells.

Fat cells store energy in the form of fats and can release this energy as heat in a process called thermogenesis. This helps to regulate the body's metabolism and prevent obesity. Vernia et al. now find that that feeding mice a high-fat diet causes widespread changes in alternative splicing in fat cells. Further bioinformatics analysis revealed that the NOVA family of splicing factors – proteins that bind to the pre-messenger RNAs to control alternative splicing – contribute to the alternative splicing of around a quarter of the genes whose splicing changes in response to a fatty diet.

Mice whose fat cells were deficient in the NOVA splicing factors displayed increased thermogenesis. As a consequence, when these animals were fed a high-fat diet, they gained less weight than animals in which NOVA proteins were present. Their metabolic activity was also better, meaning they were less likely to show the symptoms of pre-diabetes. Moreover, the activity of certain genes that are known to promote thermogenesis was greater in the fat cells that were deficient in NOVA proteins.

Overall, the results presented by Vernia et al. suggest that the normal role of NOVA proteins is to carry out an alternative splicing program that suppresses thermogenesis, which in turn may promote obesity. Drugs that are designed to target NOVA proteins and increase thermogenesis may therefore help to treat metabolic diseases and obesity. The next step is to identify the protein variants that are generated by NOVA proteins and work out how they contribute to thermogenesis.

factor for the development of diabetes and mortality (*Flegal et al., 2013*). It is therefore important that we obtain an understanding of the molecular mechanisms that control adipocyte function.

The purpose of this study was to examine alternative pre-mRNA splicing in adipocytes. We report that the consumption of a high fat diet causes differential exon inclusion/exclusion in the transcriptome. Bioinformatics analysis implicated a role for NOVA pre-mRNA splicing factors and this was confirmed by studies of mice with adipocyte-specific NOVA deficiency. Functional studies demonstrated that NOVA acts to suppress adipose tissue thermogenesis. Together, these data demonstrate that alternative pre-mRNA splicing contributes to the regulation of adipocyte biology.

## Results

### Diet-induced obesity induces a program of alternative pre-mRNA splicing

We examined gene expression in white epididymal adipocytes by RNA sequencing (Illumina Next-Seq 500 machine, 150 bp paired-end format, approximately 400 million mean reads/sample, n=3) (*Table 1*). Comparison of adipocyte mRNA isolated from mice fed (16 wk) a chow diet (CD) or a high fat diet (HFD) demonstrated differential expression of 4941 genes ($q<0.05$; absolute log2-fold change $>0.75$) and differential inclusion/exclusion of 1631 exons ($FDR<0.05$; absolute change in exon inclusion (absolute $\Delta$Inc level) $>0.1$) (corresponding to 1249 genes) in the transcriptome (*Figure 1A* and *Figure 1—figure supplements 1* and *2*). This differential exon inclusion/exclusion in mRNA most likely represents alternative pre-mRNA splicing in adipocytes. However, it is possible that some of the detected changes in pre-mRNA splicing reflect the differential presence of stromal vascular cells (*Figure 1—figure supplement 3*). Only 6.4% of the differentially expressed genes were alternatively spliced, but 25% of the alternatively spliced genes were differentially expressed (*Figure 1B*). These data indicate that the genomic response to the consumption of a HFD causes quantitative changes in both gene expression and alternative pre-mRNA splicing in adipocytes. Analysis of the co-regulated genes demonstrated enrichment for pathways including mRNA processing

**Table 1.** Summary of RNA-seq data.

| GEO Accession Subseries/ Superseries | Biological groups | Sample number | Platform | Mean read number / sample (after trimming, if applicable) | Read length (after trimming, if applicable) | Mean read alignment rate |
|---|---|---|---|---|---|---|
| GSE76294/ GSE76134 | F$^{WT}$ (3) F$^{\Delta N1}$ (3) F$^{\Delta N2}$ (3) | 9 | Illumina HiSeq 2000/ 2 x 100 bp | 135,500,000 | 100 bp | 89.4% |
| GSE76133/ GSE76134 | CD (3) HFD (3) | 6 | Illumina NextSeq 500/ 2 x 150 bp | 406,200,000 | 90 bp | 74.9% |
| GSE76317/ GSE76134 | F$^{WT}$ (4) F$^{\Delta N1,2}$ (4) | 8 | Illumina NextSeq 500/ 2 x 150 bp | 319,700,000 | 130 bp | 92.5% |

and multiple signaling pathways (e.g. insulin signaling and MAPK signaling) that are known to regulate adipose tissue biology (*Figure 1—figure supplement 4*).

The most common form of HFD-induced alternative pre-mRNA splicing was exon skipping (1052 exons), but we also detected 144 mutually exclusive exon inclusions, 160 retained introns, 123 alternative 5' splice sites, and 152 alternative 3' splice sites (*FDR*<0.05; absolute ∆Inc level >0.1) (*Figure 1C,D*).

In contrast to the extensive changes in alternative pre-mRNA splicing in adipocytes caused by diet-induced obesity (*Figure 1A–C*) few HFD-regulated alternative pre-mRNA splicing events were detected in liver, although we did find 2 skipped exons, 0 mutually exclusive exon inclusions, 13 retained introns, 1 alternative 5' splice site, and 1 alternative 3' splice site (*FDR*<0.05; absolute ∆Inc level >0.1). This comparative analysis of gene expression indicates that widespread changes in pre-mRNA splicing are not a general response to diet-induced obesity, but a selective response of white adipocytes to the consumption of a HFD.

One example of altered splicing is the mutually exclusive inclusion of exons 7a or 7b in the tyrosine kinase FYN that changes the strength of SH3 domain-mediated autoinhibition (*Brignatz et al., 2009*). Increased inclusion of *Fyn* exon 7b, compared with exon 7a, in response to the consumption of an HFD (*Figure 1—figure supplement 2B*) is anticipated to increase FYN tyrosine kinase activity (*Brignatz et al., 2009*) leading to suppression of fatty acid oxidation and promotion of insulin resistance (*Bastie et al., 2007*). Together, these data indicate that alternative pre-mRNA splicing may contribute to the adipocyte response to obesity.

## Obesity suppresses expression of the NOVA group of pre-mRNA splicing factors

To gain insight into the mechanism of HFD-induced pre-mRNA splicing, we examined exons (plus 500 bp of flanking intron sequence) to identify potential motifs that were significantly enriched for alternatively spliced exons. This analysis led to the identification of potential binding sites (YCAY) for NOVA alternative pre-mRNA splicing factors (*Darnell, 2013*). Indeed, we found significant (p<2.2 × 10$^{-16}$) enrichment of UV-mediated cross-linking and immunoprecipitation sequencing (CLIP-seq) tags for NOVA proteins identified in brain tissue (*Licatalosi et al., 2008*) within the HFD-induced group of alternatively spliced exons in adipocytes (*Figure 1E*). The NOVA CLIP-seq tags intersected with 56% of the HFD-induced alternatively spliced exons (*FDR*<0.05; absolute ∆Inc level >0.1) (*Figure 1E*) and were associated with both HFD-induced exon inclusion (53% intersection) and HFD-induced exon exclusion (62% intersection), consistent with the known role of NOVA proteins to cause context-dependent exon inclusion/exclusion (*Ule et al., 2006*). This analysis implicates a role for NOVA proteins in a sub-set of HFD-induced alternative splicing events.

NOVA proteins are expressed in several tissues, including neurons (*Darnell, 2013*), vascular endothelial cells (*Giampietro et al., 2015*), and pancreatic β cells (*Villate et al., 2014*). Whether NOVA

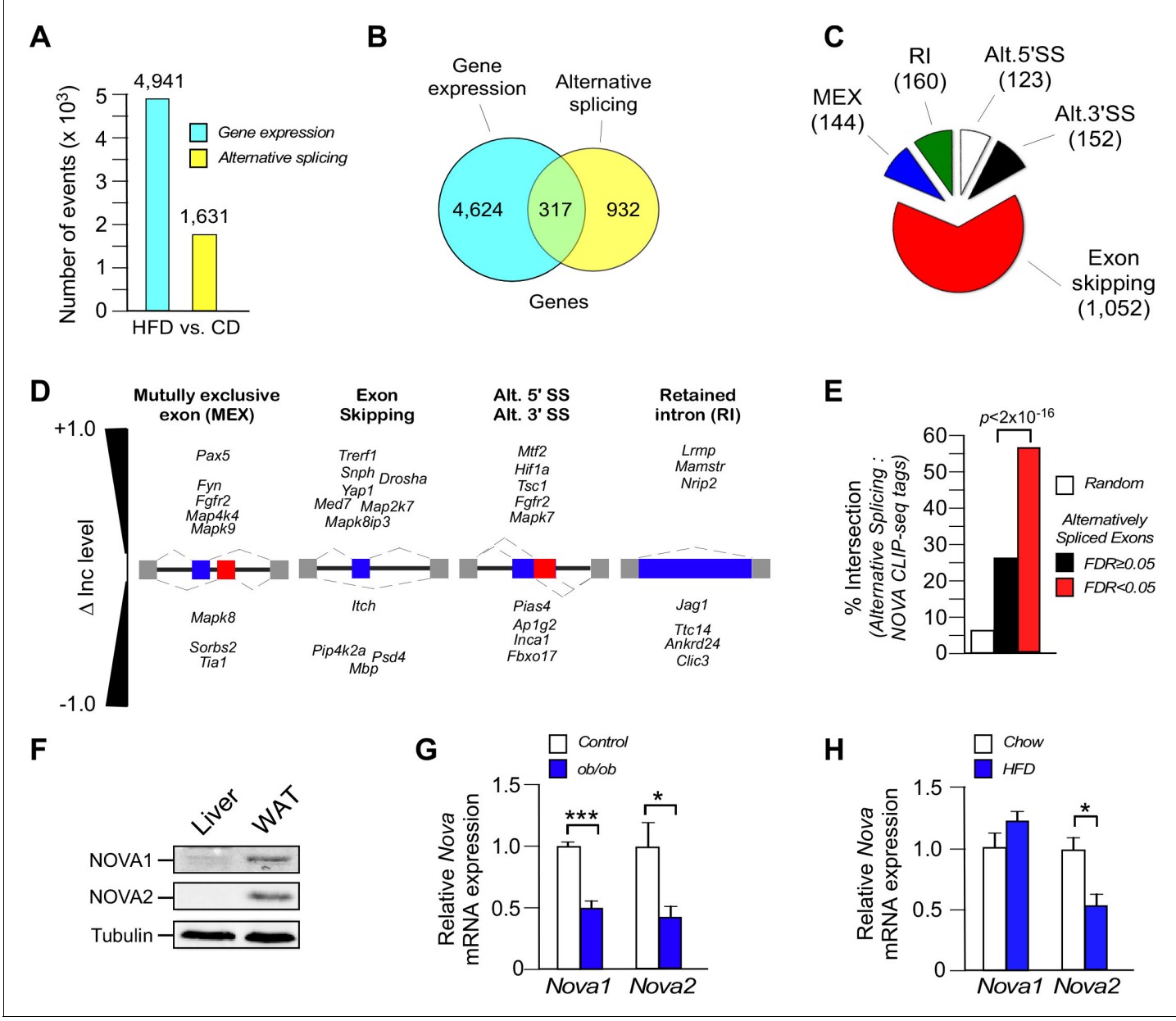

**Figure 1.** Diet-induced obesity causes changes in alternative pre-mRNA splicing in adipose tissue. (A) RNA-seq analysis demonstrates that the consumption (16 wk) of a HFD, compared with a CD, causes significant changes in mRNA expression genes ($q<0.05$; absolute log2-fold change >0.75) and differential exon inclusion/exclusion (*FDR*<0.05; absolute ΔInc level >0.1) in the epididymal adipocyte transcriptome. (B) Genes that are significantly differentially expressed and genes that are subjected to alternative pre-mRNA splicing are depicted using a Venn diagram. (C,D) Classification of HFD-induced alternative splicing events (C) and selected examples (D) are presented. (E) Enrichment of NOVA CLIP-seq tags with significantly (*FDR*<0.05; absolute ΔInc level >0.1) alternatively spliced exons (± 500 bp of intron/exon junctions) compared with non-alternatively spliced exons (*FDR*≥0.05) and random genomic sequences. (F) Immunoblot analysis of NOVA proteins in lysates prepared from hepatocytes and epididymal adipocytes. (G) *Nova1* and *Nova2* mRNA expression by epididymal adipocytes of wild-type and obese *ob/ob* mice (6 wk old) was examined by quantitative RT-PCR (mean ± SEM; n=5; *p<0.05; ***p<0.001). The source data are included as *Figure 1—source data 1*. (H) Wild-type mice were fed a CD or an HFD (16 wk). *Nova1* and *Nova2* mRNA expression by epididymal adipocytes was measured by quantitative RT-PCR analysis (mean ± SEM; n=7; *p<0.05). The source data are included as *Figure 1—source data 2*.

The following source data and figure supplements are available for figure 1:

**Source data 1.** Source data for *Figure 1G*.

**Source data 2.** Source data for *Figure 1H*.

*Figure 1 continued*

**Figure supplement 1.** Alternative pre-mRNA splicing in adipocytes.

**Figure supplement 2.** Alternative pre-mRNA splicing in adipocytes.

**Figure supplement 3.** Expression of adipocyte and stromal vascular fraction marker genes.

**Figure supplement 4.** Biological pathway enrichment analysis of differential gene expression and alternative pre-mRNA splicing caused by feeding a HFD.

**Figure supplement 5.** NOVA expression in human subcutaneous adipose tissue.

proteins are expressed in peripheral metabolic tissues is unclear. Indeed, NOVA proteins were not detected in liver, but both NOVA1 and NOVA2 proteins were found in white adipocytes (*Figure 1F*). Interestingly, *Nova* gene expression in white adipocytes was partially reduced in obese humans and mice (*Figure 1G* and *Figure 1—figure supplement 5*) and in mice fed a HFD (*Figure 1H*). This decrease in NOVA expression may be relevant to obesity-regulated changes in alternative pre-mRNA splicing in adipocytes. Moreover, the absence of NOVA expression in liver (*Figure 1F*) may contribute to the minimal effect of HFD consumption on alternative splicing of pre-mRNA in the liver.

## NOVA contributes to the obesity-induced program of alternative pre-mRNA splicing

Reduced expression of *Nova1* or *Nova2* in mice causes developmental defects and neonatal lethality (*Jensen et al., 2000*; *Ruggiu et al., 2009*). We therefore established *Nova1*$^{LoxP/LoxP}$ and *Nova2*$^{LoxP/LoxP}$ mice to study the role of NOVA proteins in adult mice with tissue-specific NOVA deficiency (*Figure 2—figure supplement 1*). *Adipoq-Cre*$^{-/+}$ mice were used to selectively ablate *Nova1* and *Nova2* genes in mature adipocytes. We initially compared control F$^{WT}$ mice (*Adipoq-Cre*$^{-/+}$) with compound mutant F$^{ΔN1,2}$ mice (*Adipoq-Cre*$^{-/+}$ *Nova1*$^{LoxP/LoxP}$ *Nova2*$^{LoxP/LoxP}$).

RNA-seq analysis of white adipocytes from HFD-fed F$^{WT}$ and F$^{ΔN1,2}$ mice (150 bp paired-end format, approximately 320 million mean reads/sample, n = 4) was performed (*Table 1*). Compound NOVA1/2 deficiency caused only a small change in gene expression (55 genes; *q*<0.05; absolute log2-fold change >0.75), but NOVA1/2 deficiency caused a large change in differential exon inclusion/exclusion (1169 exons; *FDR*<0.05; absolute Δinc level >0.1) in the adipocyte transcriptome (*Figure 2A* and *Figure 1—figure supplement 1*). These data indicate that NOVA deficiency primarily changes pre-mRNA splicing. The most common form of alternative pre-mRNA splicing caused by NOVA deficiency was exon skipping (768 exons), but we also detected 128 mutually exclusive exon inclusions, 99 intron retentions, 64 alternative 5' splice sites, and 110 alternative 3' splice sites (*Figure 2D,E*). Analysis of white adipocytes with single gene ablations of *Nova1* or *Nova2* identified fewer changes in alternative pre-mRNA splicing, including 10 & 7 skipped exons, 1 & 4 mutually exclusive exon inclusions, 12 & 16 intron retentions, 1 & 0 alternative 5' splice sites, and 2 & 4 alternative 3' splice sites, respectively (*FDR*<0.05; absolute Δinc level >0.1). These data indicate that NOVA1 and NOVA2 can cause isoform-specific changes in adipocyte pre-mRNA alternative splicing. However, the effect of compound NOVA1 plus NOVA2 deficiency to cause widespread changes in alternative pre-mRNA splicing (*Figure 2*) indicates that these NOVA proteins exhibit some functional redundancy in adipocytes.

Comparison of white adipocyte genes with differential alternative splicing (*FDR*<0.05; absolute Δinc level >0.1) caused by the consumption of a HFD (*Figure 1A*) or NOVA1/2 deficiency (*Figure 2B*) demonstrated 323 co-regulated genes (*Figure 2B*). These co-regulated genes represent 26% of the 1249 HFD-regulated genes and 34% of the 950 NOVA-regulated genes. Analysis of the co-regulated genes demonstrated enrichment for pathways including mRNA processing and multiple signaling pathways (e.g. NF-κB signaling, MAPK signaling) that contribute to the physiological regulation of adipose tissue (*Figure 2C*).

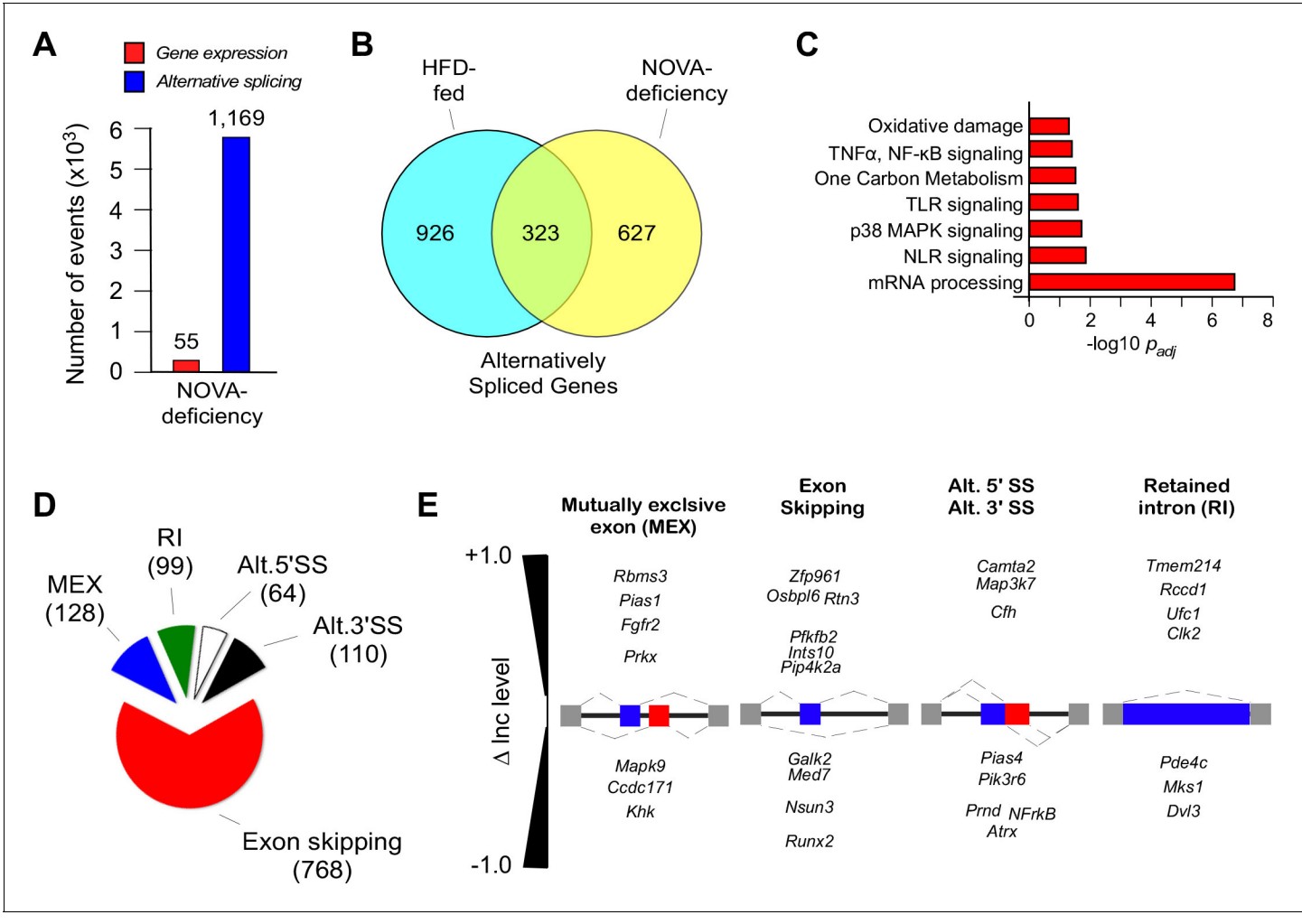

**Figure 2.** NOVA proteins contribute to alternative pre-mRNA splicing associated with diet-induced obesity. (**A**) RNA-seq analysis of F$^{WT}$ and F$^{\Delta N1,2}$ mice fed an HFD (16 wk) identifies significant changes in gene expression ($q$<0.05; absolute log2-fold change >0.75) and differential exon inclusion/ exclusion (FDR<0.05; absolute ΔInc level >0.1) in the epididymal adipocyte transcriptome. (**B**) The number of genes with significant differential alternative splicing (CD-fed vs HFD-fed WT mice and HFD-fed F$^{WT}$ mice vs HFD-fed F$^{\Delta N1,2}$ mice) are depicted using a Venn diagram. (**C**) Biological pathway enrichment analysis of the 323 genes co-regulated by alternative pre-mRNA splicing following HFD consumption and NOVA deficiency. (**D,E**) Classification of alternative splicing events caused by NOVA deficiency (**D**) and selected examples (**E**) are presented.

The following figure supplement is available for figure 2:

**Figure supplement 1.** Establishment of *Nova1$^{LoxP/LoxP}$* and *Nova2$^{LoxP/LoxP}$* mice.

## NOVA-mediated alternative pre-mRNA splicing suppresses JNK signaling in adipose tissue

To confirm the observation that NOVA proteins contribute to signaling mechanisms that can mediate metabolic regulation, we examined the expression of the JNK group of MAPK in cells that express NOVA proteins (white adipocytes) and cells that do not express NOVA proteins (hepatocytes) (*Figure 3A*). The genes *Mapk8* (encodes the JNK1 protein kinase) and *Mapk9* (encodes the JNK2 protein kinase) express pre-mRNA are alternatively spliced by the mutually exclusive inclusion of either exon 7a or 7b to yield the α and β isoforms of the JNK1 and JNK2 protein kinases (*Gupta et al., 1996*). The sequences surrounding exons 7a and 7b contain consensus sites for NOVA binding (YCAY) that are established to be NOVA binding sites by CLIP-seq analysis (*Licatalosi et al., 2008*). We designed and validated a Taqman assay to quantitate the inclusion of exon 7a or 7b sequences in *Mapk8* mRNA (*Mapk8α* and *Mapk8β*) and *Mapk9* mRNA (*Mapk9α* and *Mapk9β*)

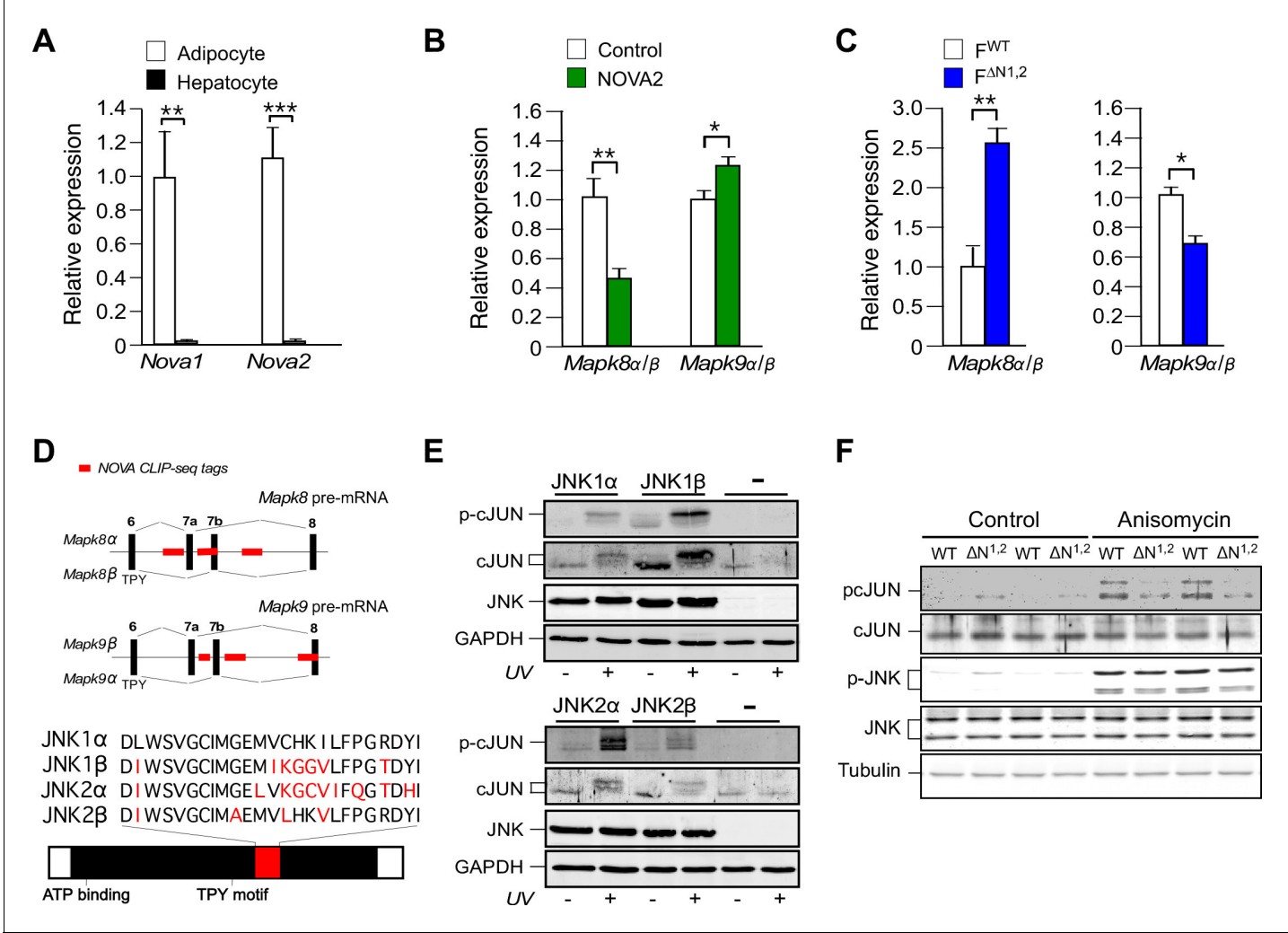

**Figure 3.** NOVA promotes signal transduction by JNK in adipose tissue. (**A**) The expression of *Nova* mRNA in adipocytes and hepatocytes was measured by quantitative RT-PCR (mean ± SEM; n=7~8; **p<0.01; ***p<0.001). The source data are included as *Figure 3—source data 1*. (**B**) The α / β expression ratios of *Mapk8* and *Mapk9* mRNA by hepatocytes was measured by quantitative RT-PCR (mean ± SEM; n=10; *p<0.05; **p<0.01). The effect of hepatic expression of GFP (Control) or NOVA2 using adenoviral vectors was examined. The source data are included as *Figure 3—source data 2*. (**C**) The expression ratio of the α and β isoforms of *Mapk8* and *Mapk9* mRNA by F^WT and F^ΔN1,2 adipocytes was measured by quantitative RT-PCR (mean ± SEM; n=5~8; *p<0.05; **p<0.01). The source data are included as *Figure 3—source data 3*. (**D**) The mutually exclusive inclusion of exons 7a or 7b in *Mapk8* and *Mapk9* mRNA is illustrated. NOVA CLIP-seq tags are highlighted in red. (**E**) *Mapk8^Δ/Δ Mapk9^-/-* MEF transduced with retroviruses expressing JNK1α, JNK1β, JNK2α, JNK2β or empty vector (-) were exposed without and with 60 J/m² UV (60 min) and lysates were examined by immunoblot analysis. (**F**) Adipocytes prepared from F^WT and F^ΔN1,2 mice were treated without and with 1 μg/ml anisomycin (10 min) and lysates were examined by immunoblot analysis.

The following source data and figure supplements are available for figure 3:

**Source data 1.** Source data for *Figure 3A*.

**Source data 2.** Source data for *Figure 3B*.

**Source data 3.** Source data for *Figure 3C*.

**Figure supplement 1.** Design and validation of Taqman assays to detect inclusion of the mutually exclusive exons 7a and 7b in *Mapk8* and *Mapk9* mRNA.

**Figure supplement 2.** Comparison of JNKα and JNKβ protein kinase activity in vitro.

*Figure 3 continued on next page*

*Figure 3 continued*

**Figure supplement 3.** Effect of adipocyte-specific JNK-deficiency on thermogenic gene expression.

(*Figure 3—figure supplement 1A,B*). This analysis demonstrated that adipocytes and hepatocytes express different alternatively spliced JNK isoforms (*Figure 3—figure supplement 1C*). These tissue-specific differences in *Mapk8/9* pre-mRNA splicing may be influenced by the selective expression of NOVA proteins in adipocytes, but not hepatocytes (*Figures 1F* and *3A*), although NOVA-independent mechanisms likely also contribute to the observed cell type-specific pattern of JNK isoform expression.

To test the role of NOVA proteins on JNK isoform expression, we examined the effect of increased and decreased NOVA expression. We found that hepatic expression of NOVA2 caused a decrease in the *Mapk8α/β* ratio and an increase in the *Mapk9α/β* ratio, indicating that NOVA can promote the expression of the *Mapk8β* and *Mapk9α* alternatively spliced isoforms (*Figure 3B*). In contrast, adipocyte-specific deficiency of NOVA1 plus NOVA2 caused an increase in the *Mapk8α/β* ratio and a decrease in the *Mapk9α/β* ratio, indicating that NOVA-deficiency promotes the expression of the *Mapk8α* and *Mapk9β* alternatively spliced isoforms (*Figure 3C*). These observations are consistent with the finding that HFD consumption causes both decreased NOVA expression (*Figure 1H*) and significant changes in the mutually exclusive inclusion of *Mapk8* exons 7a and 7b (*FDR*=0.0066). Together, these data demonstrate that NOVA proteins can promote the expression of the *Mapk8β* and *Mapk9α* isoforms.

The sequence differences between the α and β isoforms of JNK1 and JNK2 are located in the substrate binding site (*Figure 3D*) and influence the interaction of JNK with protein substrates (*Davis, 2000*). Studies using the substrate cJun demonstrate low activity (JNK1α and JNK2β) and high activity (JNK1β and JNK2α) groups of JNK protein kinases in vitro (*Figure 3—figure supplement 2*) and in vivo (*Figure 3E*). Since NOVA can promote the expression of *Mapk8β* and *Mapk9α* mRNA that encode the high activity forms of JNK1 and JNK2, NOVA-deficiency can be predicted to cause expression of the low activity forms JNK1α and JNK2β in adipocytes. To test this conclusion, we prepared primary adipocytes from $F^{WT}$ and $F^{\Delta N1,2}$ mice and examined stress-induced JNK activation. No differences in JNK expression or stress-induced activating phosphorylation ($pThr^{180}$-Pro-$pTyr^{182}$) of JNK between $F^{WT}$ and $F^{\Delta N1,2}$ adipocytes were detected by immunoblot analysis (*Figure 3F*). However, stress-induced phosphorylation of the JNK substrate $pSer^{63}$ cJun was markedly suppressed in $F^{\Delta N1,2}$ adipocytes compared with $F^{WT}$ adipocytes (*Figure 3F*). Together, these data confirm that NOVA proteins can regulate signaling mechanisms in adipocytes.

To examine the consequences of reduced JNK signaling in adipocytes, we established $F^{\Delta J1,2}$ mice with adipocyte-specific JNK-deficiency ($Adipoq-Cre^{-/+}$ $Mapk8^{LoxP/LoxP}$ $Mapk9^{LoxP/LoxP}$). These mice exhibited increased core body temperature and increased expression of genes that mediate sympathetic activation (*Adr3b*) and thermogenesis (*Ucp1*) in adipocytes (*Figure 3—figure supplement 3*). We therefore considered the possibility that NOVA-deficiency in adipocytes might also cause increased adipose tissue thermogenesis.

## NOVA-mediated alternative pre-mRNA splicing in adipose tissue regulates energy expenditure

To test the physiological role of NOVA proteins in adipocytes, we examined the effect of feeding a CD or a HFD to control $F^{WT}$ and $F^{\Delta N1,2}$ mice. We found that CD-fed $F^{WT}$ and $F^{\Delta N1,2}$ mice gained similar body mass, but HFD-fed $F^{\Delta N1,2}$ mice gained significantly less mass than HFD-fed $F^{WT}$ mice (*Figure 4A* and *Figure 4—figure supplement 1A*). Proton magnetic resonance imaging demonstrated that the decreased body mass was caused by reduced fat mass (*Figure 4B*). Analysis of organs at necropsy demonstrated a reduction in mass of the liver and adipose tissue in HFD-fed $F^{\Delta N1,2}$ mice compared with HFD-fed $F^{WT}$ mice (*Figure 4—figure supplement 1B*) and analysis of tissue sections demonstrated reduced adipocyte hypertrophy, reduced hepatic steatosis, and reduced hypertrophy of pancreatic islets in the NOVA-deficient mice (*Figure 4—figure supplement 2*).

The reduced obesity of HFD-fed $F^{\Delta N1,2}$ mice compared with $F^{WT}$ mice suggested that NOVA-deficiency may suppress HFD-induced metabolic syndrome. Indeed, hyperinsulinemia and

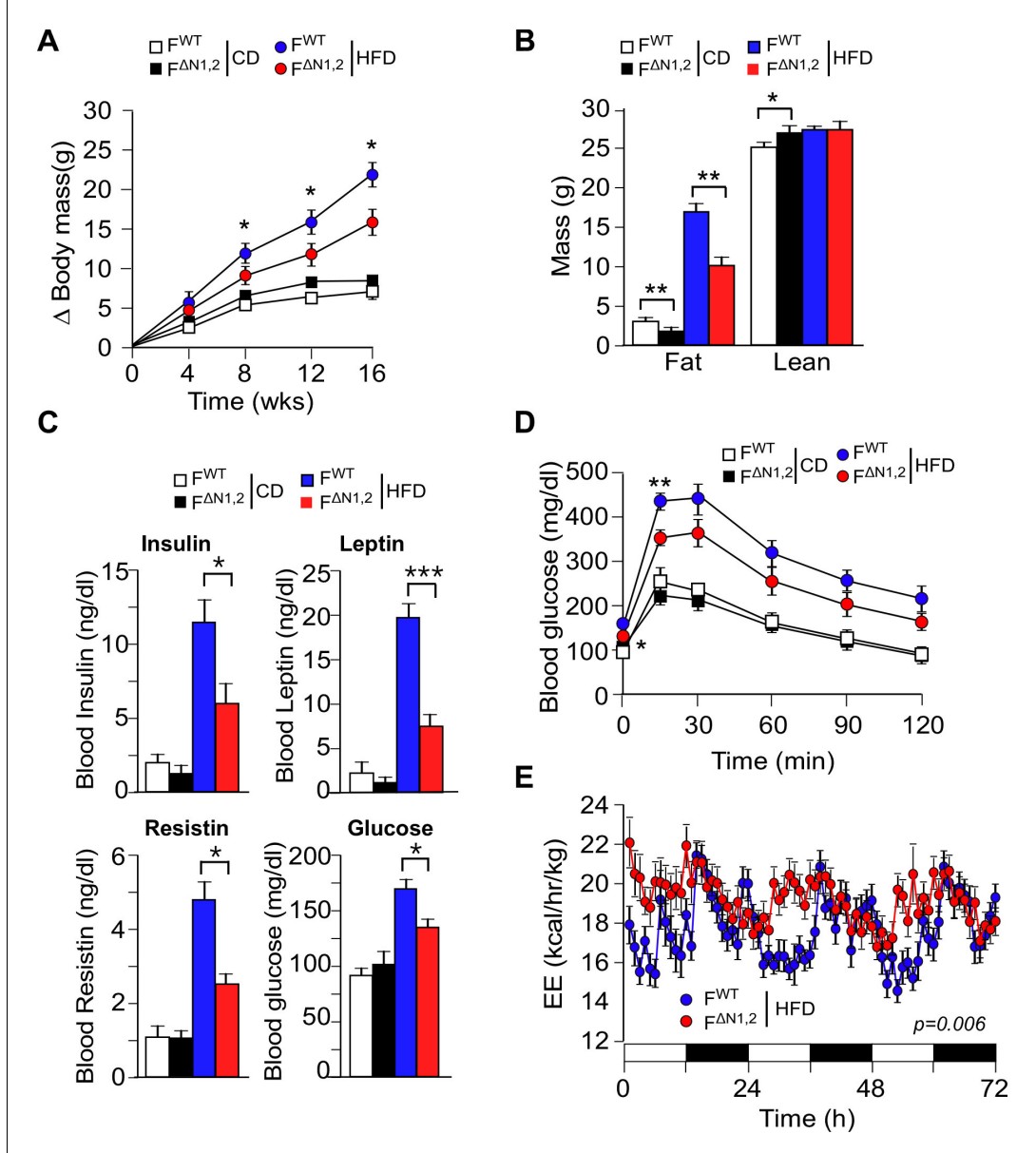

**Figure 4.** NOVA promotes the development of diet-induced obesity. (**A**) The change in body mass of $F^{WT}$ and $F^{\Delta N1,2}$ mice fed a CD or a HFD is presented (mean ± SEM; n=8~30; *p<0.05). (**B**) Body composition was examined by proton magnetic resonance spectroscopy (mean ± SEM; n=8~22; *p<0.05; **p<0.01). The source data are included as *Figure 4—source data 1*. (**C**) Blood insulin, leptin, resistin, and glucose in overnight starved CD-fed and HFD-fed (12 wks) $F^{WT}$ and $F^{\Delta N1,2}$ mice were measured (mean ± SEM; n=8~16; *p<0.05; ***p<0.001). The source data are included as *Figure 4—source data 2*. (**D**) CD-fed and HFD-fed (12 wks) $F^{WT}$ and $F^{\Delta N1,2}$ mice were examined by glucose tolerance tests (mean ± SEM; n=8~26; *p<0.05; ***p<0.001). The source data are included as *Figure 4—source data 3*. (**E**) Energy expenditure (EE) by HFD-fed (4 wks) $F^{WT}$ mice (n = 9) and $F^{\Delta N1,2}$ mice (n = 8) was examined using metabolic cages over 3 days (12 hr light; 12 hr dark). The source data are included as *Figure 4—source data 4*.

The following source data and figure supplements are available for figure 4:

**Source data 1.** Source data for *Figure 4B*.
**Source data 2.** Source data for *Figure 4C*.
**Source data 3.** Source data for *Figure 4D*.
**Source data 4.** Source data for *Figure 4E*.
*Figure 4 continued on next page*

Figure 4 continued

**Figure supplement 1.** Effect of adipose tissue-specific NOVA1/2-deficiency on organ mass.

**Figure supplement 2.** Comparison of adipose tissue, liver, and pancreas in F^WT and F^ΔN1,2 mice.

**Figure supplement 3.** Effect of adipose tissue-specific ablation of the *Nova1 or Nova2* genes.

**Figure supplement 4.** Metabolic cage analysis of physical activity and the consumption of food and water.

**Figure supplement 4—source data 1.** Source data for *Figure 4—figure supplement 4*.

**Figure supplement 5.** Metabolic cage analysis of gas exchange and energy expenditure.

**Figure supplement 5—source data 1.** Source data for *Figure 4—figure supplement 5*.

hyperleptinemia were reduced in F^ΔN1,2 mice compared with F^WT mice (*Figure 4C*). Moreover, HFD-fed F^WT mice were found to be more glucose intolerant than F^ΔN1,2 mice (*Figure 4D*) and the HFD-induced hyperglycemia observed in F^WT mice was suppressed in F^ΔN1,2 mice (*Figure 4C*). Similar (although smaller) phenotypes were detected in mice with adipocyte-specific single gene ablation of *Nova1* or *Nova2* (*Figure 4—figure supplement 3*).

Metabolic cage analysis demonstrated that food/water consumption and physical activity of F^WT and F^ΔN1,2 mice were similar (*Figure 4—figure supplement 4*), but F^ΔN1,2 mice exhibited greatly increased energy expenditure compared with F^WT mice (*Figure 4E* and *Figure 4—figure supplement 5*). This increase in energy expenditure may account for the suppression of HFD-induced obesity caused by NOVA deficiency in adipocytes. These data demonstrate that NOVA proteins in adipocytes suppress energy expenditure and promote obesity-associated metabolic syndrome.

## NOVA-mediated alternative pre-mRNA splicing regulates adipose tissue thermogenesis

We measured core body temperature by telemetry using an implanted probe in mice housed at the ambient temperature of the vivarium (21°C) and following a cold challenge (4°C). This analysis demonstrated that the core body temperature of F^ΔN1,2 mice was significantly higher than F^WT mice during the course of this study (*Figure 5A*). This was associated with increased expression of genes in sub-cutaneous adipocytes of F^ΔN1,2 mice that are associated with a 'browning' (beige/brite) phenotype (*Cidea, Dio2, Ppargc1a, Ppargc1b,* and *Ucp1*) compared with F^WT mice (*Figure 5B*). Together, these data indicate that NOVA proteins in adipocytes can suppress adipose tissue thermogenesis.

To test whether NOVA proteins act by a cell autonomous mechanism to regulate the beige/brite phenotype, we established primary adipocytes in culture. Sympathetic stimulation of adipose tissue increases cAMP and promotes the browning of white adipocytes (*Rosen and Spiegelman, 2014*). This process can be studied in vitro by treating adipocytes with drugs that raise intracellular cAMP. Indeed, treatment of primary adipocytes with the drug forskolin increases cAMP concentration and promotes the beige/brite phenotype (*Rosen and Spiegelman, 2014*), and causes increased expression of the *Adrb3, Dio2, Fgf21, Ppargc1a,* and *Ucp1* genes (*Figure 5—figure supplement 1A*). This treatment also causes decreased *Nova* gene expression and promotes expression of the alternatively spliced isoform *Mapk9β* (*Figure 5—figure supplement 1A*). To examine whether NOVA proteins contribute to the beige/brite phenotype, we compared the effects of increased cAMP on F^WT and F^ΔN1,2 primary adipocytes. This analysis demonstrated that NOVA-deficiency caused increased expression of genes in adipocytes that mediate sympathetic activation (*Adrb3*) and thermogenesis (*Ucp1*) (*Figure 5—figure supplement 1B*).

To further test the cell autonomous role of NOVA proteins on the beige/brite phenotype, we examined 3T3L1 adipocytes that are normally resistant to the effects of cAMP to promote browning. These cells express NOVA1, but NOVA2 was not detected. *Nova1* knock-down using shRNA was associated with increased expression of the beige/brite phenotype genes *Prdm16* and *Ucp1* (*Figure 5—figure supplement 2A*). Similarly, knock-down of *Mapk8β* mRNA plus *Mapk9α* mRNA using

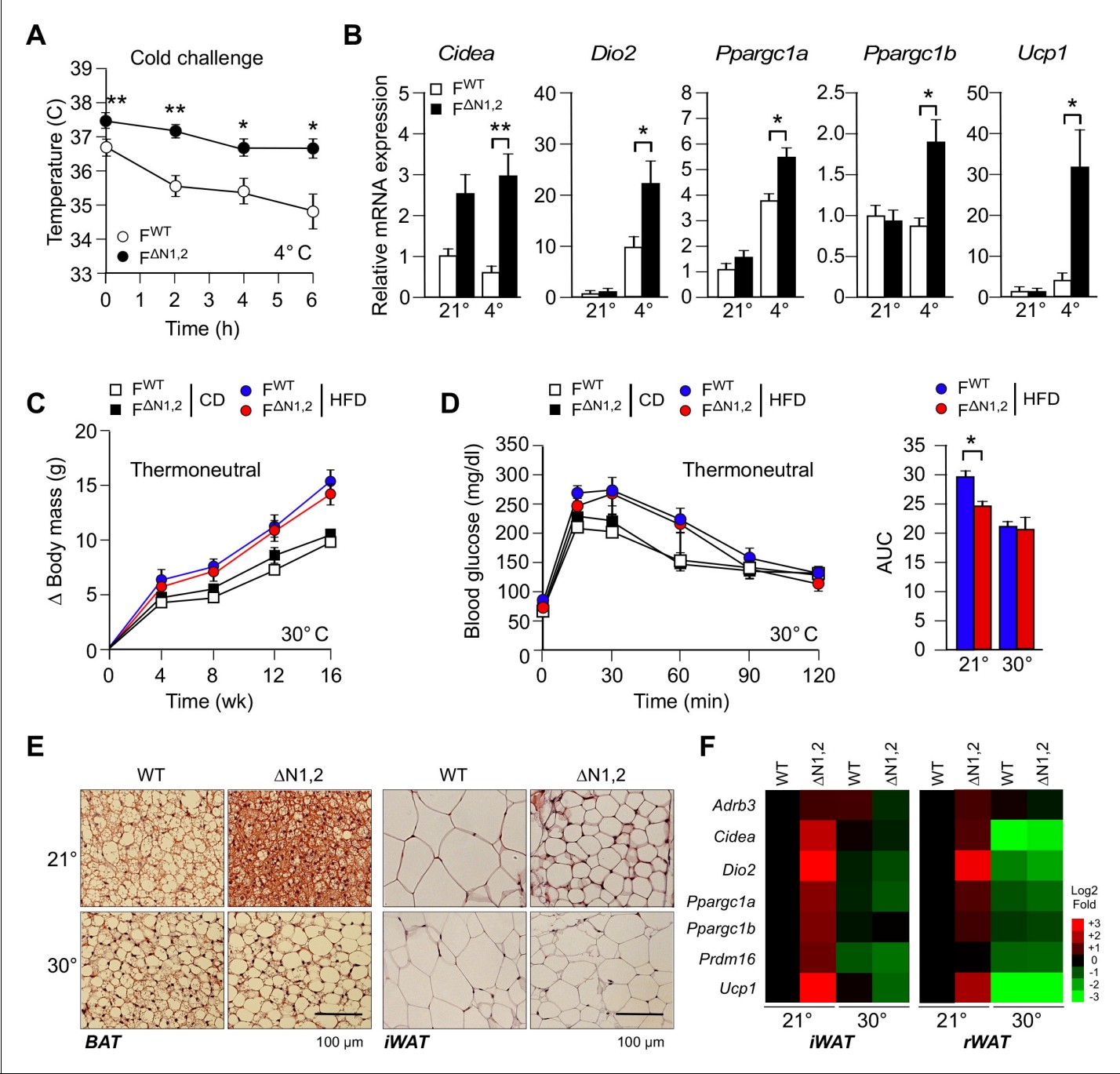

**Figure 5.** NOVA regulates a thermogenic program in adipose tissue. (A) F$^{WT}$ and F$^{\Delta N1,2}$ mice were subject to cold challenge (4°C). Core body temperature was measured by telemetry using an implanted probe (mean ± SEM; n=8; *p<0.05; **p<0.01). The source data are included as *Figure 5— source data 1*. (B) The effect of cold challenge (4°C, 6 hr) on gene expression by inguinal adipocytes (iWAT) of F$^{WT}$ and F$^{\Delta N1,2}$ mice was examined by quantitative RT-PCR (mean ± SEM; n=6~8; *p<0.05; **p<0.01). The source data are included as *Figure 5—source data 2*. (C) F$^{WT}$ and F$^{\Delta N1,2}$ mice were housed under thermal neutral conditions (30°C). The change in body mass of CD and HFD-fed mice is presented (mean ± SEM; n=7~18). (D) Glucose tolerance tests were performed on F$^{WT}$ and F$^{\Delta N1,2}$ mice housed under thermoneutral conditions (30°C). The effect of feeding a CD or a HFD (16 wk) is presented (mean ± SEM; n=8). The source data are included as *Figure 5—source data 3*. (E) Sections of brown adipose tissue (BAT) and iWAT of HFD-fed (16 wk) mice housed at 21°C and 30°C were stained with hematoxylin & eosin. The data shown are representative of 6 mice per group. (F) F$^{WT}$ and F$^{\Delta N1,2}$ mice housed at 21°C and 30°C were fed a HFD (16 wk). Gene expression by adipocytes of iWAT and retroperitoneal adipose tissue (rWAT) was examined by quantitative RT-PCR (mean, n=8). The data are presented as a heat map.

The following source data and figure supplements are available for figure 5:

*Figure 5 continued on next page*

*Figure 5 continued*

**Source data 1.** Source data for *Figure 5A*.
**Source data 2.** Source data for *Figure 5B*.
**Source data 3.** Source data for *Figure 5D*.
**Figure supplement 1.** Thermogenic gene expression by primary adipocytes.
**Figure supplement 2.** Thermogenic gene expression is increased by shRNA-mediated suppression of *Nova1* or Mapk8β/Mapk9α.

shRNA caused increased expression of the *Prdm16* and *Ucp1* genes (*Figure 5—figure supplement 2B*). Together, these data confirm that NOVA-mediated alternative splicing in adipocytes contributes to adipose tissue thermogenesis.

## NOVA-mediated adipose tissue thermogenesis contributes to the development of metabolic syndrome

To test the role of adipose tissue thermogenesis in the metabolic phenotype of $F^{\Delta N1,2}$ mice (*Figure 4*), we examined the effect of housing mice under thermoneutral conditions (30°C). We found that $F^{\Delta N1,2}$ mice and $F^{WT}$ mice gained equal body mass at thermoneutrality (*Figure 5C*). Moreover, the increased glucose tolerance of $F^{\Delta N1,2}$ mice compared with $F^{WT}$ mice at 21°C was not observed at 30°C (*Figure 5D*). These data indicate that the improved glycemic regulation exhibited by $F^{\Delta N1,2}$ mice compared with $F^{WT}$ mice at 21°C was caused by increased thermogenesis. Consistent with this conclusion, we found that the reduced adipocyte hypertrophy observed in HFD-fed $F^{\Delta N1,2}$ mice compared with $F^{WT}$ mice at 21°C was not observed at 30°C (*Figure 5E*). Similarly, the increased expression of thermogenesis-related genes (*Adrb3*, *Cidea*, *Dio2*, *Ppargc1a/b Prdm16*, and *Ucp1*) by adipocytes in HFD-fed $F^{\Delta N1,2}$ mice compared with $F^{WT}$ mice at 21°C was not found at 30°C (*Figure 5F*). Together, these data demonstrate that adipose tissue thermogenesis contributes to the improved glycemia of mice with NOVA-deficiency in adipocytes.

## Discussion

The browning of white adipose tissue is associated with the appearance of beige/brite adipocytes, increased energy expenditure, and improved obesity-induced metabolic syndrome (*Rosen and Spiegelman, 2014*). An understanding of molecular mechanisms that account for browning is therefore important for the development of potential therapies for the treatment of metabolic syndrome based on increasing adipose tissue energy expenditure. It is established that beige/brite adipocytes in white adipose tissue depots can arise from specialized progenitor cells (*Wang et al., 2013b*). Beige/brite adipocytes may also arise from inactive cells present within white adipose tissue (*Rosenwald et al., 2013*). These two mechanisms may contribute to functional beige/brite cell development (*Rosen and Spiegelman, 2014*).

The increased adipose tissue browning detected in mice with adipocyte-specific NOVA deficiency was observed using *floxed Nova* alleles and a *Cre* driver (*Adipoq-Cre*) that is expressed in mature adipocytes. It is therefore likely that the increased white adipose tissue browning caused by NOVA deficiency does not reflect a change in adipocyte differentiation from specialized progenitor cells, but rather a change in mature adipocyte function. Whether these mature adipocytes represent white adipocytes or inactive beige/brite adipocytes is unclear. However, the observation that housing mice at thermoneutrality prevents the effect of NOVA deficiency to cause 'browning' suggests that NOVA deficiency may act by promoting the activation of inactive beige/brite adipocytes.

Our analysis demonstrates that NOVA expression in white adipocytes is reduced in obese humans and mice (*Figure 1*). These changes in NOVA expression may cause changes in adipose tissue physiology. Gene knockout studies demonstrate that compound NOVA-deficiency in adipocytes caused increased adipose tissue thermogenesis and improved whole body glycemia in HFD-fed mice (*Figure 4*). However, partial NOVA-deficiency in adipocytes (potentially modeling the changes in NOVA

expression caused by feeding a HFD) caused modest changes in adipocyte physiology (*Figure 4—figure supplement 3*). The significance of the decrease in adipocyte NOVA expression in HFD-fed mice is therefore unclear. Nevertheless, our data establish that NOVA proteins in adipocytes function to suppress adipose tissue thermogenesis. The mechanism appears to be mediated by NOVA-regulated changes in pre-mRNA splicing that promote adipocyte thermogenic gene expression (*Figure 5*), but we cannot exclude additional contributions caused by other biochemical actions of NOVA proteins, including alternative mRNA polyadenylation (*Licatalosi et al., 2008*) and functional regulation of Argonaute/microRNA complexes (*Storchel et al., 2015*). Further studies will be required to examine the relative contributions of NOVA-regulated pre-mRNA splicing, alternative polyadenylation, and microRNA function. However, our analysis demonstrates that NOVA proteins do function as alternative pre-mRNA splicing factors in adipocytes (*Figure 2*).

The changes in pre-mRNA splicing caused by NOVA proteins include both increased and decreased exon inclusion, consistent with the known context-dependent role of NOVA proteins to promote both exon inclusion and exclusion (*Ule et al., 2006*). Importantly, there are significant gaps in our knowledge concerning mechanisms of alternative pre-mRNA splicing in adipocytes. First, NOVA proteins may act in a combinatorial manner with other splicing factors (*Zhang et al., 2010*), but roles for such factors in adipocytes have not been established. Second, our analysis of mice with NOVA-deficiency in adipocytes implicates a role for NOVA proteins in the alternative splicing of 26% of the genes that are regulated by HFD consumption (*Figure 2B*). Consequently, NOVA proteins do not contribute to 74% of the genes that exhibit HFD-regulated pre-mRNA splicing and mechanisms that contribute to the regulation of these genes have not been established. Further studies will be required to achieve an understanding of these processes.

The regulation of adipose tissue thermogenesis by NOVA proteins may not be entirely dependent upon the classical UCP1 pathway (*Kozak, 2010*) because of the existence of alternative thermogenic mechanisms (*Kazak et al., 2015*). Nevertheless, our data demonstrate that NOVA proteins can suppress adipocyte thermogenesis. Reduced NOVA expression in adipocytes causes thermogenesis (*Figure 5*) and may potentially be achieved by drugs targeting NOVA-mediated pre-mRNA splicing. This role of NOVA proteins is most likely mediated by a network response to a pre-mRNA splicing program that collectively regulates adipocyte thermogenesis.

In conclusion, we describe a NOVA-dependent alternative pre-mRNA splicing program in white adipocytes that regulates browning of white adipose tissue. These data identify alternative pre-mRNA splicing as a biological process that may be targeted by drugs designed to increase adipocyte thermogenesis and improve metabolic syndrome caused by obesity.

## Materials and methods

### Mice

C57BL/6J mice (RRID:IMSR_JAX:000664), B6.Cg-*Lep$^{ob}$*/J (RRID:IMSR_JAX:000632), B6;FVB-Tg(Adipoq-Cre)1Evdr (RRID:IMSR_JAX:010803) (*Eguchi et al., 2011*), B6;129-*Gt(ROSA)26Sor$^{tm1(cre/ERT)Nat}$*/J mice (RRID:IMSR_JAX:004847) (*Badea et al., 2003*), and B6.129S4-*Gt(ROSA)26Sor$^{tm1(FLP1)Dym}$*/RainJ mice (RRID:IMSR_JAX:009086) (*Farley et al., 2000*) were obtained from The Jackson Laboratory. We have previously reported *Mapk8$^{LoxP/LoxP}$* (*Das et al., 2007*), *Mapk9$^{LoxP/LoxP}$* mice (*Han et al., 2013b*), and *Mapk9$^{-/-}$* mice (RRID:IMSR_JAX:004321) (*Yang et al., 1998*).

We generated *Nova1* and *Nova2* conditional mice using ES cells targeted by homologous recombination (*Nova1$^{tm1a(EUCOMM)Hmgu}$* and *Nova2$^{tm1a(KOMP)Wtsi}$*), the preparation of chimeric mice, and breeding to obtain germ-line transmission of the disrupted *Nova1* and *Nova2* genes using standard methods. The *Frt-Neo$^R$-Frt* cassette was excised by crossing with *FLPeR* mice to obtain mice with the *Nova1$^{LoxP}$* and *Nova2$^{LoxP}$* alleles. Mice were genotyped by PCR analysis of genomic DNA using the primers 10F (5'-GTCCGTAAGGCATGTC-3') and 2R (5'-AGCAAAAAGCCATCCATG-3') to detect the *Nova1$^+$* (894 bp), *Nova1$^{LoxP}$* (1,101 bp), and *Nova1$^\Delta$* (281 bp) alleles, the primers 1F (5'-CAGAAGAACTGGAGAC-3') and N2-2R (5'- GGTTGGGCTGTCAGTG-3') to detect the *Nova2$^+$* (149 bp) and *Nova2$^{LoxP}$* (127 bp) alleles, or with the primers N2delF1 (5'-CAGGCTGGCGCCGGAAC-3') and N2-2R (5'-GGTTGGGCTGTCAGTG-3') to detect the *Nova2$^{LoxP}$* (970 bp) and *Nova2$^\Delta$* (153 bp) alleles.

All mice were studied on the C57BL/6J strain background. Male mice (8 wks old) were fed a chow diet (Iso Pro 3000, Purina) or a HFD (S3282, Bioserve). Body weight was measured with a scale. Whole body fat and lean mass were non-invasively measured using $^1$H-MRS (Echo Medical Systems).

The mice were housed at 21°C (alternatively at 4°C or 30°C, where indicated) in a specific pathogen-free facility accredited by the American Association for Laboratory Animal Care (AALAC). The Institutional Animal Care and Use Committee (IACUC) of the University of Massachusetts Medical School approved all studies using animals.

## Metabolic cages

The analysis was performed by the Mouse Metabolic Phenotyping Center at the University of Massachusetts Medical School. The mice were housed under controlled temperature and lighting with free access to food and water. The food/water intake, energy expenditure, respiratory exchange ratio, and physical activity were measured using metabolic cages (TSE Systems).

## Body temperature

Biocompatible and sterile microchip transponders (IPTT-300 Extended Accuracy Calibration; Bio Medic Data Systems) were implanted subcutaneously. Cold tolerance tests (4°C) were performed using mice fed a chow diet *ad-libitum*.

## Blood analysis

Blood glucose was measured with an Ascensia Breeze 2 glucometer (Bayer). Adipokines and insulin in plasma were measured by multiplexed ELISA using a Luminex 200 machine (Millipore).

## Glucose and insulin tolerance tests

Glucose and insulin tolerance tests were performed by intraperitoneal injection of mice with glucose (1 g/kg) or insulin (1.5 U/kg) using methods described previously (*Sabio et al., 2008*).

## Hepatic expression of NOVA

Mice (8 wks) were fed a HFD. At 12 wks, the mice treated with $5 \times 10^9$ pfu/mouse Adenovirus-NOVA2 or Adenovirus-GFP (Applied Biological Materials Inc.) by tail vein injection. The mice were euthanized at 2 wks post-injection.

## Plasmids

Human Flag-tagged *Mapk8α1*, *Mapk8β1*, *Mapk9α2*, and *Mapk9β2* cDNA cloned in the expression vector pCDNA3 have been described previously (*Gupta et al., 1996*). Murine *Mapk8α1*, *Mapk8β1*, *Mapk9α2*, and *Mapk9β2* cDNA were isolated by RT-PCR and cloned by blunt-end ligation in the *Sna*B1 site of the retroviral expression vector pBABE-puro (Addgene plasmid #1764) (*Morgenstern and Land, 1990*). Retroviral stocks were prepared by transfection of Phoenix-ECO cells (American Type-Culture Collection, ATCC CRL-3214) (*Lamb et al., 2003*).

The DNA sequences used to generate shRNA vectors were *Nova1* (5'-CCGG<u>GCTGCTCAGTA</u><u>TTTAATTA</u>CTCGAGTAATTAAATACTGAGCAGCTTTTTG-3'), *Mapk8β* (5'-CCGG<u>TCATGGGAGAAA</u><u>TGATCAAAG</u>CTCGAGCTTTGATCATTTCTCCCATGATTTTTG-3'), *Mapk9α* (5'-CCGG<u>GTGAAAGG</u><u>TTGTGTGATATTC</u>CTCGAGGAATATCACACAACCTTTCACTTTTTG-3'). These sequences were cloned in the *Age1/EcoR1* sites of the lentiviral vector pLKO.1-puro (Addgene #8453; [*Stewart et al., 2003*]). Lentiviral stocks were prepared by transfection of 293T cells with the indicated replication-incompetent lentiviral vector (pLKO1-shRNA) together with the packaging plasmid psPAX2 and the envelope plasmid pMD2.G (Addgene #12259 and #12260; [*Naldini et al., 1996*]).

## Adipocyte tissue culture

Primary inguinal adipocytes were prepared from male mice (8 wk old). The fat pads were minced with a razor blade and incubated (40 min, 37°C) in 12.5 mM Hepes pH 7.4, 120 mM NaCl, 6 mM KCl, 1.2 mM MgSO$_4$, 1 mM CaCl$_2$, 2%BSA, 2.5 mM glucose, 1 mg/ml collagenase II (Sigma) and 0.2 mg/ml DNAse I (Sigma). The digested tissue was filtered through a 100 µm nylon strainer and centrifuged (8 min at 250 g). The pellet was suspended in Dulbecco's modified Eagle's medium (DMEM):Ham's F12 (1:1) medium with 8%FBS, 1x Antibiotic-Antimycotic, 2 mM glutamine (Life

Technologies). 100,000 cells/well were seeded in 12 well plates. The medium was refreshed every 2 days. On day 6, differentiation was induced using medium further supplemented with 0.5 mM IBMX (Sigma), 5 µg/ml insulin (Sigma), 1 µM Troglitazone (TZD; Calbiochem), and 2.5 µM dexamethasone (Sigma). On day 8, the medium was refreshed using medium supplemented with insulin and troglitazone only. On day 11, the medium was refreshed using medium supplemented with insulin only. Mature adipocytes were observed on day 14.

3T3-L1 MBX cells (American Type-Culture Collection, ATCC CRL-3242) were cultured in high glucose DMEM supplemented with 10% fetal bovine serum (FBS), sodium pyruvate (1 mM), glutamine (2 mM), and 100 units/ml penicillin, and 100 µg/ml streptomycin (Life Technologies). Transduction assays were performed using pLKO1 lentiviruses and selection with 2 µg/ml puromycin. The cells were differentiated to adipocytes by growing to confluence for 48–72 hr. On day 0, the media were changed to media supplemented with 0.5 mM IBMX (Sigma), 5 µg/ml insulin (Sigma), 1 µM Troglitazone (TZD; Calbiochem), and 1 µM dexamethasone (Sigma)). This medium was refreshed every 2 days. On day 4, the medium was refreshed using medium supplemented with insulin and TZD only. On day 6, the medium was refreshed using medium supplemented with 0.5 µg/ml insulin only. Mature adipocytes were observed on day 8.

## Murine embryo fibroblasts

E13.5 primary murine fibroblasts (MEF) obtained from mice that express 4-hydroxytamoxifen-inducible *Cre* were established in culture (*Das et al., 2007*). $Cre^{ERT2-/+}$ $Mapk8^{+/+}$ $Mapk9^{+/+}$ MEF and $Cre^{ERT2-/+}$ $Mapk8^{LoxP/LoxP}$ $Mapk9^{-/-}$ MEF (*Das et al., 2007*) were cultured in DMEM supplemented with 10% fetal bovine serum, 100 units/ml penicillin, 100 µg/ml streptomycin, 0.1 mM 2-mercaptoethanol, and 2 mM L-glutamine (Life Technologies). Transduction assays were performed using pBABE-puro retroviruses and selection with 2 µg/ml puromycin (*Lamb et al., 2003*). Cells were treated with 1 µM 4-hydroxytamoxifen (24h) and subsequently cultured (5 days). The cells were exposed without or with 60 J/m$^2$ UV-C and incubated (60 mins) prior to harvesting.

## Protein kinase assays

Lysates were prepared from TNFα-treated (10 ng/ml, 10 mins) and non-treated transfected HEK 293T/17 cells (American Type-Culture Collection, ATCC CRL-11268) expressing Flag-tagged JNK proteins (*Gupta et al., 1996*). JNK proteins were isolated by immunoprecipitation using agarose-bound Flag M2 antibody (Sigma-Aldrich Cat# A2220, RRID:AB_10063035) (*Gupta et al., 1996*). JNK activity was measured using an in vitro protein kinase assay with the substrates cJun and [γ-$^{32}$P]ATP as substrates (*Whitmarsh and Davis, 2001*).

## Immunoblot analysis

Tissue extracts were prepared using Triton lysis buffer (20 mM Tris-pH 7.4, 1% Triton-X100, 10% glycerol, 137 mM NaCl, 2 mM EDTA, 25 mM β-glycerophosphate, 1 mM sodium orthovanadate, 1 mM PMSF and 10 µg/mL leupeptin plus aprotinin). Extracts (30–50 µg of protein) were examined by immunoblot analysis by probing with antibodies to cJUN (Cell Signaling Technology Cat# 9165L, RRID:AB_2129578), pSer$^{63}$-cJUN (Cell Signaling Technology Cat# 9261S, RRID:AB_2130162), Flag M2 (Sigma-Aldrich Cat# F1804, RRID:AB_262044), GAPDH (Santa Cruz Biotechnology Cat# sc-365062, RRID:AB_10847862), JNK1/2 (BD Biosciences Cat# 554285, RRID:AB_395344), pThr$^{180}$-Pro-pTyr$^{182}$-JNK (pJNK) (Cell Signaling Technology Cat# 4668P, RRID:AB_10831195), NOVA1 (Abcam Cat# ab97368, RRID:AB_10680798), NOVA2 (Sigma-Aldrich Cat# AV40400, RRID:AB_1854572), and βTubulin (BioLegend Cat# 903401, RRID:AB_2565030). Immunocomplexes were detected by fluorescence using anti-mouse and anti-rabbit secondary IRDye antibodies (Li-Cor) and quantitated using the Li-Cor Imaging system

## Analysis of tissue sections

Histology was performed using tissue fixed in 10% formalin for 24 hr, dehydrated, and embedded in paraffin. Sections (7 µm) were cut and stained using hematoxylin & eosin (American Master Tech Scientific). Paraffin sections were stained with an antibody to insulin (Dako Cat# A0564, RRID:AB_10013624) that was detected by incubation with anti-Ig conjugated to Alexa Fluor 488 (Life

Technologies). DNA was detected by staining with DAPI (Life Technologies). Fluorescence was visualized using a Leica TCS SP2 confocal microscope equipped with a 405-nm diode laser.

## Adipocyte isolation

Inguinal, retroperitoneal, and epididymal fat pads were surgically removed at necropsy. Adipocytes were isolated after incubation (40 min at 37°C with shaking) of adipose tissue in 12.5 mM Hepes pH 7.4, 120 mM NaCl, 6 mM KCl, 1.2 mM $MgSO_4$, 1 mM $CaCl_2$, 2% BSA, 2.5 mM glucose, 1 mg/ml collagenase II (Sigma #C6885) and 0.2 mg/ml DNAse I (Sigma #DN25)). Larger particles were removed using a 100 μm nylon sieve and the filtrates were centrifuged at 1000 rpm (3 min). Floating adipocytes were washed twice with 1x PBS and subsequently centrifuged at 1000 rpm (3 min) prior to RNA isolation. The stromal vascular fraction (SVF) was collected after centrifugation at 3000 rpm (5 min) prior to RNA isolation. The expression of adipocyte marker genes (*Adipoq* & *Leptin*), SVF marker genes (*Emr1 (F4/80)*, *Itgam (Cd11b)*, *Cd68*) in adipocytes and the SVF was measured by quantitative RT-PCR analysis (*Figure 1—figure supplement 3*). The expression of a gene (*Fabp4*) that is expressed by both adipocytes and SVF was also examined.

## RNA sequencing

RNA was isolated using the RNeasy kit (Qiagen). RNA quality (RIN > 9) was verified using a Bioanalyzer 2100 System (Agilent Technologies). Total RNA (10 μg) was used for the preparation of each RNA-seq library by following the manufacturer's instructions (Illumina). Three or four independent libraries prepared from different mice were sequenced (Illumina) for each condition. *Table 1* presents a summary of the adipocyte RNA-seq data and associated GEO accession numbers. The liver RNA-seq data (CD vs HFD (n=3)) were previously reported (*Vernia et al., 2014*) (GEO accession number GSE55190).

## Quantitative RT-PCR analysis of gene expression

The expression of mRNA was examined by quantitative PCR analysis using a Quantstudio PCR system (Life Technologies). Taqman assays were used to quantitate *Adipoq* (Mm00456425_m1), *Adrb3* (Mm02601819_g1), *Cd68* (Mm03047340_m1), *Dio2* (Mm00515664_m1), *Emr1* (F4/80) (Mm00802530_m1), *Fabp4* (Mm00445880_m1), *Itgam* (Cd11b) (Mm00434455_m1), *Leptin* (Mm00434759_m1), *Mapk8* (Mm00489514_m1), *Mapk9* (Mm00444231_m1), *Nova1* (Mm01289097_m1), *Nova2* (Mm01324153_m1), *Ppargc1a* (Mm00447183_m1), *Ppargc1b* (Mm00504720_m1), *Prdm16* (Mm00712556_m1), *Ucp1* (Mm01244861_m1) mRNA and *18S* RNA (4308329) (Applied Biosystems). Standard curves were constructed using the threshold cycle (Ct) values for each template dilution plotted as a function of the logarithm of the amount of input template. The number of mRNA copies for each gene-sample combination was calculated using the slope of the standard curve. To obtain a normalized abundance, copy numbers were corrected for the amount of *18S* RNA in each sample.

## Quantitative RT-PCR analysis of alternative splicing

The inclusion of exons 7a and 7b in *Mapk8* and *Mapk9* mRNA was examined by quantitative PCR using the Quantifast probe PCR kit (Qiagen) and the following combination of primers and Taqman probes (Applied Biosystems): *Mapk8α*: Fwd, GGAGAACGTGGACTTATGGTCTGT; Probe: 6FAM-TGCCACAAAATCCT-MGBNFQ; Rev, TGATCAATATAGTCCCTTCCTGGAA. *Mapk8β*: Fwd, GAACGTTGACATTTGGTCAGTTG; Probe, 6FAM-AGAAATGATCAAAGGTGGTGTT-MGBNFQ; Rev, TCAATATGATCTGTACCTGGGAACA. *Mapk9α*: Fwd, GGTCAGTGGGTTGCATCATG; Probe, 6FAM-AGCTGGTGAAAGGTT-MGBNFQ; Rev, TGATCAATATGGTCAGTACCTTGGA. *Mapk9β*: Fwd, ATCTGGTCTGTCGGGTGCAT; Probe: 6FAM-AAATGGTCCTCCATAAAG-MGBNFQ; Rev, GATCAATATAGTCTCTTCCTGGGAACA. *Mapk8* and *Mapk9* spliced variants were quantitated using the relative quantification method. The alternative splicing is represented as the ratio *Mapk8α/Mapk8β* and *Mapk9α/Mapk9β*.

## Semi-quantitative RT-PCR analysis of alternative splicing

RT-PCR analysis was performed using amplimers based on the sequences of *Adam15* exons 19 and 21 (svADAM15F1: 5'-GCGGGCACAGCAGATGAC-3' and svADAM15R1: 5'- GGGTTGGCAGGCAG

TGGC-3′) and *Yap1* exons 5 and 7 (svYap1F1: 5-GGAGAAGGAGAGACTG-3′ and svYap1R2: 5′-GTCCCTCCATCCTGCTC-3′). The PCR products were examined by agarose gel electrophoresis and staining with ethidium bromide. *Adam15* mRNA and exon 20-skipped *Adam15* mRNA were detected as 277 bp and 78 bp DNA fragments. *Yap1* mRNA and exon 6-skipped *Yap1* mRNA were detected as 144 bp and 96 bp DNA fragments.

## Bioinformatics analysis

Fastqc v0.10.1 (http://www.bioinformatics.babraham.ac.uk/projects/fastqc/) and cutadapt (v1.7) (https://pypi.python.org/pypi/cutadapt/1.7.1) were used to generate sequence quality reports and trim/filter the sequences respectively. Reads below a minimum quality PHRED score of 30 at the 3′ end were trimmed (*Table 1*). The filtered reads were aligned to the mouse reference genome (Ensembl GRCm38). Alignments were carried out using Bowtie2 (v2.2.1.0) (*Langmead and Salzberg, 2012*) and Tophat2 (v2.0.9) (*Kim et al., 2013*). Samtools (v0.0.19) (*Li et al., 2009*) and IGV (v2.3.60) (*Thorvaldsdottir et al., 2013*) were used for indexing the alignment files and viewing the aligned reads respectively. Gene expression was quantitated as fragments per kilobase of exon model per million mapped fragments (FPKM) using Cufflinks (v2.2.0) (*Trapnell et al., 2010*). Differentially expressed genes were identified using Cufflinks tools (Cuffmerge and Cuffdiff). CummeRbund (v2.4.1) (*Trapnell et al., 2012*) was used to assess biological replicate concordance. Significant changes in gene expression were defined as $q<0.05$ and absolute log2 fold change >0.75. Gene sets were examined using Wikipathways in Webgestalt (*Wang et al., 2013a*). Alternative splicing was examined using rMATS software (v3.0.9) (*Shen et al., 2014*). rMATS was run using default settings to compute *p*-values and *FDR*s of splicing events. Significant changes in alternative splicing were defined as $FDR<0.05$ and absolute ΔInc level >0.1. Rmats2sashimiplot (https://github.com/Xinglab/rmats2sashimiplot) and Sashimiplot (*Katz et al., 2015*) were used for quantitative visualization of alternative exon expression from rMATS.

The microarray analysis of human adipose tissue mRNA expression data (GEO GSE25402) (*Arner et al., 2012*) was performed using Genespring. The summarization method was based on RMA16, the normalization method was based on the median approach on log2 scale, and the fold change was computed from average signal intensity values.

NOVA interactions with pre-mRNA were examined using CLIP-seq data obtained from murine brain (*Licatalosi et al., 2008*) provided (in BED format) by Dr. R. B. Darnell (Rockefeller University). The coordinates were based on the mm9 assembly. The UCSC liftOver utility was used to convert the NOVA CLIP-seq tag coordinates from the mm9-based assembly to the mm10/GRCm38 assembly. The coordinates for the NOVA CLIP-seq tags were sorted based on chromosome and start position. Custom PERL scripts (*Source code 1* and *2*) were used to extract three groups of genomic regions: (1) statistically significant differentially expressed alternatively spliced exons (*FDR*<0.05 and the absolute inclusion level difference >0.1) (n = 1631); (2) exons not meeting statistical significance (*FDR*≥0.05) (n = 11,490); and (3) randomly selected regions of the mouse genome comprising 100 bp (n = 10,000). The coordinates for each genomic region were expanded to include an additional 500 bp of sequence flanking the 5′ and 3′ ends of the genomic region. If an exon belongs to a gene comprising one or more alternatively spliced exons with *FDR*<0.5 and one or more alternatively spliced exons with *FDR*≥0.05, the exon was excluded from the second group. In each group, the regions were sorted based on the chromosome and the start position. Duplicated regions were removed. The BEDtools (2.22.0) intersect command (*Quinlan, 2014*) was used to determine the intersection between the regions in each group and the NOVA-CLIP-seq tags. The number of intersecting regions between a group and the NOVA binding sites were tallied. For each group, the number and the percentage of regions with and without NOVA binding sites were calculated (*Figure 1E*). Thus, NOVA CLIP-tags intersecting with the following groups were examined: (1) statistically significant differentially spliced exons plus 500 bp of flanking sequence both upstream and downstream; (2) non-alternatively spliced exons plus 500 bp of flanking sequence both upstream and downstream; (3) random genomic sequences of similar fragment size (*Figure 1E*). Statistical significance between two groups was determined by the Pearson's Chi-squared test.

## Statistical analysis of experimental data

Values are given as means ± SEM of at least three independent experiments. Sample sizes were determined by prior experimentation. Differences between groups were examined for statistical significance using the Student´s test or analysis of variance (ANOVA) with the Fisher's test (*p<0.05, **p<0.01 and ***p<0.001).

## Acknowledgements

We thank Robert Darnell for providing NOVA CLIP-seq data, Armanda Roy for expert technical assistance, and Kathy Gemme for administrative assistance. The RNA-seq data has been deposited in the Gene Expression Omnibus (GEO) database with accession number GSE76134. RJD is an investigator of the Howard Hughes Medical Institute. This study was supported by NIH grant R01 DK107220 (to RJD). The UMASS Mouse Metabolic Phenotyping Center is supported by NIH grant UC2 DK093000 (to JKK).

## Additional information

### Competing interests

RJD: Reviewing editor, *eLife*. The other authors declare that no competing interests exist.

### Funding

| Funder | Grant reference number | Author |
|---|---|---|
| National Institute of Diabetes and Digestive and Kidney Diseases | UC2 DK093000 | Jason K Kim |
| National Institute of Diabetes and Digestive and Kidney Diseases | R01 DK107220 | Roger J Davis |

The funders had no role in study design, data collection and interpretation, or the decision to submit the work for publication.

### Author contributions

SV, Conception and design, Acquisition of data, Analysis and interpretation of data, Drafting or revising the article; YJKE, Analysis and interpretation of data, Drafting or revising the article; MSH, Acquisition of data, Analysis and interpretation of data, Drafting or revising the article; JC-K, TB, JKK, Acquisition of data, Analysis and interpretation of data; RJD, Conception and design, Analysis and interpretation of data, Drafting or revising the article

### Author ORCIDs

Roger J Davis, http://orcid.org/0000-0002-0130-1652

### Ethics

Human subjects: The reported human studies are limited to the analysis of de-identified human gene expression data previously deposited in the GEO database (accession no: GSE25402). This analysis is exempt from a requirement for specific ethical approval.

Animal experimentation: This study was performed in strict accordance with the recommendations in the Guide for the Care and Use of Laboratory Animals of the National Institutes of Health. All of the animals were handled according to approved institutional animal care and use committee (IACUC) protocols (#A-1032) of the University of Massachusetts Medical School.

## Additional files

### Supplementary files

• Source code 1. Custom Perl Script 1.

• Source code 2. Custom Perl Script 2.

**Major datasets**

The following dataset was generated:

| Author(s) | Year | Dataset title | Dataset URL | Database, license, and accessibility information |
|---|---|---|---|---|
| Vernia S, Edwards YJK, Han MS, Cavanagh-Kyros J, Barrett T, Kim JK, Davis RJ | 2016 | A pre-mRNA alternative splicing program promotes adipose tissue thermogenesis | http://www.ncbi.nlm.nih.gov/geo/query/acc.cgi?acc=GSE76134 | Publicly available at the NCBI Gene Expression Omnibus (accession no: GSE76134). |

The following previously published datasets were used:

| Author(s) | Year | Dataset title | Dataset URL | Database, license, and accessibility information |
|---|---|---|---|---|
| Arner E, Mejhert N, Kulyte A, Balwierz PJ, Pachkov M, Cormont M, Lorente-Cerbrian S, Ehrlund A, Laurencikiene J, Heden P, Dahlman-Wright K, Tanti JF, Hayashizaki Y, Ryden M, Dahlman I, van Nimwegen E, Daub CO, Arner P | 2012 | Adipose tissue microRNAs as regulators of CCL2 production in human obesity | http://www.ncbi.nlm.nih.gov/geo/query/acc.cgi?acc=GSE25402 | Publicly available at the NCBI Gene Expression Omnibus (accession no: GSE25402). |
| Vernia S, Cavanagh-Kyros J, Garcia-Haro L, Sabio G, Barrett T, Jung DY, Kim JK, Shula HP, Garber M, Gao G, Davis RJ | 2014 | The PPARalpha-FGF21 hormone axis contributes to metabolic regulation by the hepatic JNK signaling pathway | http://www.ncbi.nlm.nih.gov/geo/query/acc.cgi?acc=GSE55190 | Publicly available at the NCBI Gene Expression Omnibus (accession no: GSE55190). |

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
