## [Decision Letter]

Thank you for submitting your article "A pre-mRNA Alternative Splicing Program Promotes Adipose Tissue Thermogenesis" for consideration by *eLife*. Your article has been favorably evaluated by Fiona Watt (Senior editor) and three reviewers, one of whom is a member of our Board of Reviewing Editors. The reviewers have opted to remain anonymous.

The reviewers have discussed the reviews with one another and the Reviewing Editor has drafted this decision to help you prepare a revised submission.

The authors have investigated the contribution of alternative splicing to the development of diet-induced obesity. They observe obesity-induced changes in programs of adipocyte gene expression and alternative splicing associated with multiple signalling pathways that are known to regulate adipose tissue biology. They link changes in alternative splicing to Nova splicing regulators through an analysis of motif enrichment, published CLIP-Seq data, and the surprising observation that Nova proteins are expressed in adipocytes. The authors also provide evidence that Nova deficiency in adipocytes results in increased adipose tissue thermogenesis and improved glycemia. Collectively, the authors' study represents an important advance in understanding the role of alternative splicing in metabolic regulation and its contribution to the development of diet-induced obesity. The authors are requested to address the following points in a revised manuscript.

1) The methods suggest that adipocytes were isolated by flotation before RNA-Seq was performed, but the results just say "adipose tissue mRNA". The distinction is crucial, because events that are attributed to alternative splicing could reflect different levels of specific isoforms in cell types that differ in number over the course of a high fat diet. Please describe the isolation protocol in more detail, and show some evidence that isolation was successful and close to complete, e.g. macrophage gene expression in the stromal-vascular fraction, and adipocyte-specific genes (e.g. leptin, adiponectin) in the fat cell fraction.

2) More information on the motif enrichment analysis should be provided. How was this analysis performed and were YCAY motifs the most highly enriched in association with obesity induced changes in alternative splicing? If other enriched motifs were detected (which is typically the case with such analyses), what are they? While it is not the authors' responsibility to comprehensively identify splicing pathway in adipocytes, some perspective on the present results would help a great deal, especially since many of the alternative splicing events lack NOVA binding sites.

3) In Figure 1 and Figure 2 the authors report that approximately 6,000 alternative splicing events change between normal diet and HFD, as well as for WT and NOVA1/2 mutants on HFD. However, the ΔPSI cut-off used is >1%. In most studies employing RNA-Seq analysis to analyze splicing changes, ΔPSI cut-offs applied are typically >10%, in large part because smaller ΔPSI values are unreliable (they are within the noise range of the detection) and unlikely biologically significant. This raises some concern about the CLIP-Seq analysis in which ~42% of the HFD induced AS events are claimed to have NOVA CLIP-Seq tags (Figure 1). What was the ΔPSI cut-off used for this analysis? Do the authors observe similar results (NOVA motif and GO term enrichment etc.) when a more stringent ΔPSI is applied? In general, RNA-Seq measurements of alternative splicing should be validated by performing RT-PCR assays on a representative subset of the detected changes (see also comment 6 below).

4) The authors make use of Nova CLIP-Seq datasets from the Darnell lab, which were generated from brain tissue. This is not apparent unless one digs into the Methods section. Ideally, the authors should confirm Nova binding to pre-mRNA sequences associated with Nova- regulated exons in adipocytes, or at the very least mention in the main text the source of their data and the caveats associated with comparing binding data from another tissue source. Keeping in mind point 3 above, it would be informative to know whether enriched YCAY motifs supported by CLIP-Seq data form distributions that are predictive of increased versus decreased exon inclusion (as per the binding maps published by Darnell and colleagues), when Nova is expressed in versus depleted from adipocytes.

5) Subheading: "NOVA-mediated alternative pre-mRNA splicing regulates adipose tissue thermogenesis". Although this statement is supported by the experiments examining *JNK* alternative splicing, the authors cannot exclude that much of the biology they observe may arise from other Nova-dependent changes. In this regard, what is the extent of overlap in mRNA level changes when comparing CD-fed vs HFD-fed WT mice and HFD-fed FWT mice vs HFD-fed F∆N1,2 mice? Can the authors confirm whether NOVA-mediated splicing of JnK1/2 inhibits thermogenic gene expression program by restoring the splicing of JnK1/2 in 3T3L1 adipocytes?

6) In Figure 3—figure supplement 1 expression of *Jnk1e, 1* and *Jnk2, 2* splice isoform ratios in adipocytes (express *Nova1/2*) and hepatocytes (do not express *Nova1/2*) is calculated using qPCR assays. From the figure it seems that *Nova1/2* promote the expression of *Jnk1m* and *Jnk2.* This is contradictory to what Figure 3 show. Is this due to incorrect α/β ratio calculation? The legend in Figure 3—figure supplement 1 states that α/β ratio is calculated; however, the numbers on the graph show the β/α ratio. A comparison of *Jnk1/2* isoform ratios in two different tissues after normalizations further convolutes the results. Also, it is recommended that the authors use a different approach to quantify *Jnk1/2* splice isoform ratios as qPCR assays can give inaccurate results due to different primer efficiencies between separate assays. Use of semi-quantitative, end-point RT-PCR assays (with primers to flanking constitutive exons so as to simultaneously detect included and skipped isoforms) is quite standard in the field.

7) In Figure 1, reduced *Nova1/2* mRNA levels are shown in ob/ob mice compared to controls. But ob/ob is a genetic model of obesity. Are *Nova1/2* mRNA and protein levels affected with diet-induced obesity? NOVA1 and NOVA2 western blots should be performed on normal-fed and HFD-fed adipocytes to strengthen the claims.

8) How do the authors reconcile the observations that NOVA levels go down in over-nutrition and obesity, but loss of NOVA in fat promotes thermogenesis and energy expenditure? Is the thought that the diet-induced reduction in NOVA is a compensatory act to prevent massive weight gain?

[Editors' note: further revisions were requested prior to acceptance, as described below.]

Thank you for resubmitting your work entitled "A pre-mRNA Alternative Splicing Program Promotes Adipose Tissue Thermogenesis" for further consideration at *eLife*. Your revised article has been favorably evaluated by Fiona Watt as the Senior editor and Ben Blencowe as the Reviewing editor.

It is recommended that the authors modify their title ahead of publication to: "A Nova-Regulated Alternative Splicing Program Associated with Adipose Tissue Thermogenesis", or simply by removing 'pre-mRNA' in their current title. The authors should also correct the following sentence in the Abstract by inserting 'a' as indicated: 'Together, these data provide ‘a’ quantitative analysis of gene expression at exon-level resolution in obesity'.

---

## [Author Response]

*1) The methods suggest that adipocytes were isolated by flotation before RNA-Seq was performed, but the results just say "adipose tissue mRNA". The distinction is crucial, because events that are attributed to alternative splicing could reflect different levels of specific isoforms in cell types that differ in number over the course of a high fat diet. Please describe the isolation protocol in more detail, and show some evidence that isolation was successful and close to complete, e.g. macrophage gene expression in the stromal-vascular fraction, and adipocyte-specific genes (e.g. leptin, adiponectin) in the fat cell fraction.*

We have revised the manuscript to expand the description of the procedure employed to isolate adipocytes, subsection “Adipocyte isolation”, and we have included new figure to document macrophage and adipocyte gene expression in isolated adipocytes and the stromal vascular fraction (Figure 1—figure supplement 3).

*2) More information on the motif enrichment analysis should be provided. How was this analysis performed and were YCAY motifs the most highly enriched in association with obesity induced changes in alternative splicing? If other enriched motifs were detected (which is typically the case with such analyses), what are they? While it is not the authors' responsibility to comprehensively identify splicing pathway in adipocytes, some perspective on the present results would help a great deal, especially since many of the alternative splicing events lack NOVA binding sites.*

We have revised the manuscript to provide a detailed description of the methods used to associate NOVA motifs with HFD-induced alternative pre-mRNA splicing in the subsection “Bioinformatics analysis”. We find an intersection of NOVA motifs with 56% of HFD-induced alternatively spliced exons (Figure 1). However, functional analysis using the adipocyte-specific NOVA1/2 knockout mice implicates NOVA in the alternative pre-mRNA splicing of only 26% of the HFD-regulated genes (Figure 2). As noted by the editor and the reviewers, this means that the majority (74%) of HFD-induced gene splicing events are NOVA- independent. Our analysis does not provide insight into these NOVA-independent splicing events. Tackling this problem is a key goal for our future studies.

*3) In Figure 1 and Figure 2 the authors report that approximately 6,000 alternative splicing events change between normal diet and HFD, as well as for WT and NOVA1/2 mutants on HFD. However, the ΔPSI cut-off used is >1%. In most studies employing RNA-Seq analysis to analyze splicing changes, ΔPSI cut-offs applied are typically >10%, in large part because smaller ΔPSI values are unreliable (they are within the noise range of the detection) and unlikely biologically significant. This raises some concern about the CLIP-Seq analysis in which ~42% of the HFD induced AS events are claimed to have NOVA CLIP-Seq tags (Figure 1). What was the ΔPSI cut-off used for this analysis? Do the authors observe similar results (NOVA motif and GO term enrichment etc.) when a more stringent ΔPSI is applied? In general, RNA-Seq measurements of alternative splicing should be validated by performing RT-PCR assays on a representative subset of the detected changes (see also comment 6 below).*

We are very grateful to the reviewers for making this point. We agree that a more stringent threshold for exon inclusion/exclusion will most likely provide increased biological significance. To achieve better transparency for our analysis, we have included a new supplementary Figure that shows the number of alternative splicing events (for both HFD and NOVA KO adipocytes) at different FDR and ∆ Inc levels (Figure 1—figure supplement 1). We have also revised Figure 1 to present all the RNA-seq data with increased threshold stringencies: a) RNA-seq differential gene expression, q<0.05; log2-fold change >0.75; and b) RNA-seq differential exon inclusion/exclusion, FDR<0.05, absolute ∆Inc level >0.1. We believe that this change represents a substantial improvement in the presentation of our data.

The analysis of the intersection of NOVA CLIP-seq tags with the HFD-induced alternatively spliced exons presented in revised Figure 1 was performed using the more stringent thresholds (FDR<0.05, absolute ∆Inc level >0.1). The analysis shows 56% intersection (p<2x10^-16^).

The GO analysis (Figure 1—figure supplement 4 & Figure 2) presented in the revised manuscript is based on the more stringent thresholds for differential gene expression (q<0.05; log2-fold change >0.75;) and alternative splicing (FDR<0.05, absolute ∆Inc level >0.1).

We have revised the manuscript to include semi-quantitiative RT-PCR analysis of the effects of HFD and NOVA-deficiency on selected genes (Figure 1—figure supplement 2). These data complement the quantitative RT-PCR analysis of splicing presented in Figure 3 and Figure 3—figure supplement 1 and Figure 5—figure supplement 1.

*4) The authors make use of Nova CLIP-Seq datasets from the Darnell lab, which were generated from brain tissue. This is not apparent unless one digs into the Methods section. Ideally, the authors should confirm Nova binding to pre-mRNA sequences associated with Nova- regulated exons in adipocytes, or at the very least mention in the main text the source of their data and the caveats associated with comparing binding data from another tissue source. Keeping in mind point 3 above, it would be informative to know whether enriched YCAY motifs supported by CLIP-Seq data form distributions that are predictive of increased versus decreased exon inclusion (as per the binding maps published by Darnell and colleagues), when Nova is expressed in versus depleted from adipocytes.*

Thank you for raising this important point. We have revised the manuscript in the first paragraph of the subsection “Obesity suppresses expression of the NOVA group of pre-mRNA splicing factors” to explicitly state that the CLIP-seq tag data were obtained from brain tissue rather than the relevant cell type (adipocytes). Clearly the use of adipocyte CLIP-seq data would represent a significant improvement in our analysis. However, we have not yet been able to successfully adapt CLIP protocols to the analysis of adipocytes, most likely because of the lipid-laden nature of these cells. We will need to develop new procedures before we will able to include CLIP studies in our analysis of adipocytes.

We have examined the intersection of NOVA CLIP-seq tags with exons that are differentially included/excluded in our analysis (FDR<0.05, absolute ∆Inc level >0.1). The intersection is 53% for included exons and 62% for excluded exons in adipocytes of ice fed a HFD. The analysis is consistent with the RNA map reported by Darnell and colleagues (Ule et al. (2006) Nature 444, 580-586). We have revised the manuscript to include this information in the aforementioned subsection.

*5) Subheading: "NOVA-mediated alternative pre-mRNA splicing regulates adipose tissue thermogenesis". Although this statement is supported by the experiments examining JNK alternative splicing, the authors cannot exclude that much of the biology they observe may arise from other Nova-dependent changes. In this regard, what is the extent of overlap in mRNA level changes when comparing CD-fed vs HFD-fed WT mice and HFD-fed FWT mice vs HFD-fed F∆N1,2 mice? Can the authors confirm whether NOVA-mediated splicing of JnK1/2 inhibits thermogenic gene expression program by restoring the splicing of JnK1/2 in 3T3L1 adipocytes?*

Thank you for this comment concerning the Discussion section of our manuscript. We agree. Our data do support a role for NOVA-mediated pre-mRNA splicing, but we cannot exclude a role for other possible functions of NOVA proteins. Examples include alternative mRNA polyadenylation (Licatalosi et al. 2008) and functional regulation of Argonaute/microRNA complexes (Storchel et al. 2015). The manuscript has been revised to explicitly state this in the third paragraph of the Discussion.

There are 4,941 significant changes in gene expression caused by HFD (Figure 1) and 55 significant changes in gene expression caused by NOVA-deficiency in HFD-fed mice (Figure 1) (q<0.05; log2-fold change ≥0.75). The comparison requested indicates that there is little similarity between the very large change in gene expression caused by HFD (4,941 genes) and the modest change in gene expression caused by NOVA- deficiency (55 genes) in HFD-fed mice.

The proposed complementation assay using JNK is an interesting experiment, although rather artificial using 3T3 L1 cells. We are currently tackling this problem using mice engineered to conditionally express in a tissue-specific manner exon 7a or 7b *Jnk1* and *Jnk2*. We believe that this study, which is beyond the scope of the present analysis, will provide a definitive complementation test for the role of JNK. In the present study, we show that JNK is a target of NOVA-mediated splicing and that JNK signaling is repressed by NOVA. This analysis therefore validates the conclusion that NOVA can regulate signaling in adipocytes. However, the overall conclusions of our study – stated in the Discussion section of the manuscript is that “this role of NOVA proteins is most likely mediated by a network response to a pre-mRNA splicing program that collectively regulates adipose tissue thermogenesis”. We believe that it is premature to focus on a specific NOVA target (e.g. JNK) when we know that 950 genes exhibit NOVA-mediated changes in splicing (Figure 2). For this reason, “JNK” is not mentioned in the Abstract or the Discussion sections of the manuscript.

*6) In Figure 3—figure supplement 1 expression of Jnk1α, 1β and Jnk2α, 2β splice isoform ratios in adipocytes (express Nova1/2) and hepatocytes (do not express Nova1/2) is calculated using qPCR assays. From the figure it seems that Nova1/2 promote the expression of Jnk1α and Jnk2β. This is contradictory to what Figure 3 show. Is this due to incorrect α/β ratio calculation? The legend in Figure 3—figure supplement 1 states that α/β ratio is calculated; however, the numbers on the graph show the β/α ratio. A comparison of Jnk1/2 isoform ratios in two different tissues after normalizations further convolutes the results. Also, it is recommended that the authors use a different approach to quantify Jnk1/2 splice isoform ratios as qPCR assays can give inaccurate results due to different primer efficiencies between separate assays. Use of semi-quantitative, end-point RT-PCR assays (with primers to flanking constitutive exons so as to simultaneously detect included and skipped isoforms) is quite standard in the field.*

We agree with the point noted concerning the *Jnk* splice forms expressed in liver and adipocytes. These tissue-specific differences in splicing most likely reflect non-NOVA regulatory mechanisms that influence the splicing of JNK1 and JNK2. This is an important point that is now stated in the revised manuscript in the first paragraph of the subsection “NOVA-mediated alternative pre-mRNA splicing suppresses JNK signaling in adipose tissue”. Nevertheless, our conclusion that NOVA proteins promote the expression of JNK1β & JNK2α and repress the expression of JNK1α & JNK2β is supported by two observations. First, expression of NOVA2 in the liver causes a decrease in the JNK1α/β ratio and an increase in the JNK2β/α ratio (Figure 3). Second, NOVA1/2-deficiency in adipocytes causes an increase in the JNK1α/β ratio and a decrease in the JNK2α/β ratio (Figure 3). Together, these data demonstrate that NOVA proteins promote the expression of JNK1β & JNK2α. Nevertheless, liver and adipocytes clearly exhibit tissue-specific differences in JNK expression that are NOVA-independent. Our current studies are focused on understanding this tissue-specific different in *Jnk1* and *Jnk2* pre-mRNA alternative splicing. Studies of this NOVA-independent mechanism fall outside of the scope of the present analysis.

Thank you for identifying the error in Figure 3—figure supplement 1. You are correct, the β/α ratio was presented instead of the stated α/β ratio. This has now been corrected in revised Figure 3—figure supplement 1.

Semi-quantitative RT-PCR assays are useful. Indeed, we have performed such assays to validate some changes in alternative pre-mRNA splicing (Figure 1—figure supplement 2). However, this approach is not appropriate for studies of the mutually exclusive inclusion of exons 7a and 7b in *Jnk1* and *Jnk2* using flanking primers to exons 6 & 8. This is because the length of exons 7a and 7b is identical. Consequently, exon 6 & 8 primers in a semi-quantitative PCR assay cannot distinguish between the inclusion of exons 7a or 7b. It is for this reason that we developed a quantitative Taqman assay using exon 6 & 8 primers to measure the inclusion of exons 7a and 7b in *Jnk1* and *Jnk2* mRNA. The method is described in detail in the manuscript in the subsections “Quantitative RT-PCR analysis of gene expression” and “Quantitative RT-PCR analysis of alternative splicing” and is fully validated (Figure 3—figure supplement 1).

*7) In Figure 1, reduced Nova1/2 mRNA levels are shown in ob/ob mice compared to controls. But ob/ob is a genetic model of obesity. Are Nova1/2 mRNA and protein levels affected with diet-induced obesity?*

We have revised the manuscript to present data on NOVA expression in CD-fed and HFD-fed mice (Figure 1).

*8) How do the authors reconcile the observations that NOVA levels go down in over-nutrition and obesity, but loss of NOVA in fat promotes thermogenesis and energy expenditure? Is the thought that the diet-induced reduction in NOVA is a compensatory act to prevent massive weight gain?*

It is most likely that the changes in NOVA expression represent a compensatory mechanism. However, it should be noted that the obesity-induced decrease in NOVA expression is relatively modest ~ 50%. This modest decrease in NOVA expression should be interpreted in the context of our knockout mouse analysis that demonstrates minor phenotypes for adipocyte-specific NOVA1 knockout mice and NOVA2 knockout mice (Figure 4—figure supplement 3). The large effect of NOVA-deficiency on thermogenesis was only detected in the compound NOVA1/2 knockout mice (Figure 4). This analysis of mutant mice suggests that the decrease in NOVA expression caused by obesity is likely to lead to only modest changes in adipose tissue thermogenesis.

However, we cannot exclude the possibility that the HFD-induced decrease in NOVA expression plays a key role in other aspects of adipocyte biology.

[Editors' note: further revisions were requested prior to acceptance, as described below.]

It is recommended that the authors modify their title ahead of publication to: "A Nova-Regulated Alternative Splicing Program Associated with Adipose Tissue Thermogenesis", or simply by removing 'pre-mRNA' in their current title. The authors should also correct the following sentence in the Abstract by inserting 'a' as indicated: 'Together, these data provide <a> quantitative analysis of gene expression at exon-level resolution in obesity'.

We have revised the title of the paper by deleting the words “pre-mRNA”, as requested. The revised title is: “An Alternative Splicing Program Promotes Adipose Tissue Thermogenesis”. We have also corrected the Abstract, as requested.